

**Reviews and Synthesis: To the bottom of carbon processing at**
**the seafloor**
**Jack J. Middelburg***
Earth Sciences, Utrecht University, PO Box 80 021, 3508 TA Utrecht, The
Netherlands
*\* Invited contribution by Jack J. Middelburg, recipient of the EGU Vladimir*
*Vernadsky Medal 2016*
**Abstract**
Organic carbon processing at the seafloor is studied by geologists to
better understand the sedimentary record, by biogeochemists to quantify burial
and respiration, by organic geochemists to elucidate compositional changes and
by ecologists to follow carbon transfers within food webs. Here I review these
disciplinary approaches and discuss where they agree and disagree. It shown
that the biogeochemical approach (ignoring the identity of organisms) and the
ecological approach (focussing on growth and biomass of organisms) are
consistent on longer time scales. It is hypothesized that secondary production by
microbes and animals might impact the composition of sedimentary organic
matter eventually buried. Animals impact sediment organic carbon processing
by microbes in multiple ways: by governing organic carbon supply to sediments
and by mixing labile organic matter to deeper layers. An inverted microbial loop
is presented in which microbes profit from bioturbation rather than animals
profiting from microbial processing of otherwise lost dissolved organic
resources. Sediments devoid of fauna therefore function differently and are less
efficient in processing organic matter with the consequence that more organic
matter is buried and transferred from Vernadsky's biosphere to the geosphere.
**1 Introduction**
The seawater-sediment interface represents one of the largest interfaces
on earth and our knowledge of processes at and fluxes through this dynamic and
understudied interface is rather limited. This interface extends a few cm-dm
upwards into the water column, i.e. benthic boundary layer (Boudreau and
Jørgensen, 1992), as well as few cm-dm into the sediments, i.e. the bioturbated,
active surface layer (Berner, 1980; Meysman et al., 2006; Aller, 2013). It serves
as a habitat for organisms, governs the partitioning of material being buried or
recycled and acts as a filter for the paleorecord (Rhoads, 1974). Processes in the
surface sediment layer determine whether carbon is recycled within the
biosphere (short-term cycle) or transferred to the geosphere (long-term cycle)
and as such it functions as a key interface in the System Earth.
This pivotal role of the seafloor in processing material deposited has been
studied by scientists from various disciplines with their own interests,
techniques and paradigms (Figure 1). Marine geologists and paleoceanographers

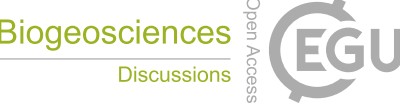



study sediments with the primary aim to extract information on past
environmental conditions using down-core measurements of substances
delivered to the seafloor and that have survived the processing at the seafloor
(Burdige, 2006; Bender, 2013). Biogeochemists quantify the fate of material
delivered, in particular how much of the material is eventually buried or
processed, and determine when and in what form the remaining part is recycled
to the water column, because recycling of key nutrients (e.g. N, P, Si, Fe) sustains
primary production (Berner, 1980; Aller, 1980, 2001, 2013; Soetaert et al.,
2000). Organic geochemists investigate how organic matter delivered to the
seafloor is degraded, transformed or preserved using changes in the composition
at the molecular level (Hedges and Keil, 1995; Dauwe et al, 1999; Burdige, 2006;
Bianchi and Canuel, 2011). Ecologists focus on the organisms, i.e. the actors
consuming, producing and transporting the material deposited (Gage and Tyler,
1992; Gray and Elliot, 2009; Herman et al., 1999; Krumins et al., 2013).
Although these disciplines often study the same topic, e.g. organic matter
delivered to the seafloor, they focus on different aspects and usually ignore key
concepts, findings and approaches from other disciplines. For example,
ecologists and biogeochemist studying carbon flows at the seafloor normally
ignore detailed molecular information available from organic geochemistry.
Bioturbation, particle transport and mixing by animals at the seafloor , is often
ignored by paleoceanographers, and biogeochemists have developed advanced
transport-reaction models in which the actors, the animals, mix the particles but
do so without consuming organic matter, their food.
Here I present the existing views on organic carbon processing at the
seafloor, discuss where they agree and disagree and aim to arrive at an
integrated view of carbon processing at the seafloor that is consistent with
recent views within the organic geochemical, sediment geochemical, ecological
and microbiological research communities.  This overview is necessarily and
admittedly incomplete but rather covers personal interests and presents new
concepts on this topic. It is a concise version of the Vernadsky Medal Lecture
presented at the 2017 EGU meeting.
**2 Geochemists focus on quantification of burial and mineralization**.
Organic matter delivered to marine sediments eventually is either
mineralized and the metabolites (carbon dioxide and nutrients) accumulate in
pore-waters and exchange with the overlying water, or it is not consumed and
then buried through the steady accumulation of particles (Fig. 2). This
geochemical view is highly simplified, but for that reason also quantitative and
instructive (Berner, 1980; Boudreau, 1997; Aller, 2013). The percentage of
organic matter buried varies from less than one % to a few tens % and is closely
related with total sediment accumulation rate (Canfield, 1989, 1994; Middelburg
et al., 1993). Since sediment accumulation rates are much higher in vegetated,
coastal, deltaic, shelf and ocean-margin settings, the majority of organic matter
burial occurs there, with organic carbon burial in deep-sea sediments accounting
for <5 %  (Berner, 1982; Duarte et al., 2005; Burdige, 2007). In the deep sea and
other low burial efficiency settings, almost all organic matter is degraded to
inorganic carbon and organic matter degradation rates provide an excellent
proxy for organic matter deposition (Jørgensen, 1982; Cai and Reimers, 1995;




Glud, 2008). Organic matter degradation can be quantified via the consumption
of oxygen, the production of dissolved inorganic carbon and through the use of
pore-water data and diagenetic models (i.e. reaction-transport models for
sediments). At steady state sediment oxygen consumption provides an accurate
measure for total sediment organic matter degradation, independent whether
organic matter is degraded aerobically (i.e. with oxygen) or anaerobically (with
alternative electron acceptors such as nitrate, metal oxides, sulphate), because
almost all reduced metabolites released (ammonium, manganese (II), iron(II),
hydrogen sulfide and methane) are re-oxidized (Jørgensen, 1977, 2006; Berner
and Westrich, 1985; Aller and Rude, 1988; Soetaert et al., 1996; Boetius et al.,
2000; Strous and Jetten, 2004; Raghoebarsing et al., 2006; Middelburg and Levin,
2009).
Geochemists have adopted a transport-reaction modeling approach to
accurately quantify organic matter processing (Berner, 1980; Boudreau, 1997;
Burdige, 2006). The basic premise of these diagenetic models is that both
particles and solutes are subject to transport and reaction, making them distinct
from for instance groundwater transport-reaction models in which normally
only solutes and gas phases are mobile. Transport of solutes is due to molecular
diffusion, pore-water advection and biologically mediated processes: enhanced
diffusion due to interstitial fauna (Aller and Aller, 1992) and bio-irrigation due
tube and burrow construction and flushing by macrofauna (Aller, 1980, 1984;
2001). Particle transport is not only due to steady particle deposition but also
due to animal activities (bioturbation, Boudreau, 1997; Aller, 1994, 2013; Rice,
1986; Meysman et al., 2003, 2006, 2010). The reaction terms in these diagenetic
models are normally limited to microbial and chemical reactions and described
using zero, first, second order or Monod/Michaelis-Menten type kinetics
(Bouldin, 1968; Berner, 1980; Soetaert et al., 1996; Boudreau, 1997). There is a
major inconsistency in the basic conceptual model underlying the (numerical)
diagenetic models: animals dominate transport processes via pore-water
irrigation and particle mixing, but without consuming any organic matter. This is
has not received much attention because the ruling paradigm within the
microbial ecological and geochemical research communities is that animals
contribute very little to total carbon processing. Multiple recent studies involving
use of $^{13}$C as deliberate tracers show that this premise does not hold on the short
term (days to weeks; Blair et al., 1996; Moodley et al., 2002, 2005a; Woulds et al.,
2009, 2016). Another reason for continuing with the simple approach is that our
understanding of particle mixing due to animals is very limited (Meysman et al.,
2006; 2010), that particle movement may require little energy because of
fracturing (Dorgan et al., 2005) and that diagenetic models can very accurately
reproduce most observations (Soetaert et al., 1996; Berg et al., 2003).
These models combined with solid-phase and pore-water concentration
vs. depth profiles, sediment-water exchange fluxes and rate measurement have
resulted in a consistent picture of organic matter degradation pathways in
marine sediments (Berner, 1980; Boudreau, 1997; Aller, 2013). These models
can predict where, when and why organic matter oxidation occurs aerobically or
involves nitrate, metal oxides or sulphate as oxidants (Rabouille and Gaillard,
2001; Boudreau, 1996; Soetaert et al., 1996; Middelburg et al. 1996; van
Cappellen and Wang, 1996; Archer et al., 2002; Meysman et al., 2003; Berg et al.,
2003). They also resolve the re-oxidation of reduced products such as





ammonium, manganese(II), iron (II), sulfide and methane (Fig. 2) and as such
define the scope for aerobic and anaerobic organisms and the distribution and
activity of chemoautotrophs. Chemoautotrophs in sediments produce about 0.4
Tg C y$^{-1}$, similar to the riverine delivery of organic carbon to the ocean
(Middelburg, 2011). However, these diagenetic models cannot predict organic
carbon burial rates, nor do they provide much insight why organic matter is
buried or why it is labile (reactive) or rather refractory. For this we need to have
a detailed look at the organic geochemistry of sediment organic carbon.
**3 Organic geochemists focus on the composition of organic matter**
**preserved**
Organic matter delivered to the seafloor is predominantly produced in the
surface sunlit layer of the ocean (Fig. 3). This organic matter is rich in proteins,
carbohydrates and lipids and follows Redfield stoichiometry (Bianchi and
Canuel, 2011). Organic matter processing leads to preferential degradation of the
more labile components with the result that organic matter becomes less
reactive (Middelburg, 1989; Arndt et al., 2013) and organic matter composition
changes (Fig. 1; Wakeham et al. 1997; Dauwe et al., 1999; Lee et al., 2000). The
proportion of organic matter that can be characterized molecularly decreases
with progressive degradation, i.e. with water depth or depth downcore
(Wakeham et al., 1997; Hedges et al., 2000; Middelburg et al., 1999; Nierop et al.,
2017). This molecularly uncharacterizable material increases to more than 70%
in deep-sea organic matter. The organic geochemical approach to study organic
matter processing is limited not only by our inabilities to characterize the
majority of the sedimentary organic matter, but also by the simple fact that the
degraded fraction cannot be studied and we have to base our knowledge on the
small fraction of extensively processed organic material remaining.
The changes in organic matter composition due to organic matter
processing have been utilized to estimate the lability-digestability or its reverse
the refractory nature of organic matter with various proxies such as chlorophyll
to bulk organic matter, fraction of nitrogen present as amino acid, and the
contribution of proteins and carbohydrates to total organic matter (Cowie et al.,
1992; Dell'Anno et al., 2000; Danovaro et al., 2001; Koho et al., 2013). The amino-
acid based degradation index (Dauwe and Middelburg, 1998) is one of the most
commonly used proxies to quantify the extent of degradation or the quality of
the remaining particulate organic matter (Dauwe et al., 1999; Keil et al., 2000).
The compositional changes have also been used to infer transformation of
organic matter by bacteria (production of bacterial transformation products;
accumulation of D-amino acids; Cowie and Hedges, 1994; Dauwe et al., 1999;
Grutters et al., 2001; Vandewiele et al., 2009; Lomstein et al., 2006, 2012), extent
of degradation under oxic and anoxic conditions (Sinninghe Damsté et al., 2002;
Huguet et al, 2008; Nierop et al., 2017) and the relative importance of bacterial
and faunal pathways of organic matter degradation pathways (Sun et al., 1999;
Woulds et al., 2012, 2014). Although some organic geochemical studies hint at
the importance of secondary production (new organic matter produced by
microbes and animals and other heterotrophs; Cowie and Hedges, 1994; Grutters
et al., 2001; Lomstein et al., 2006, 2012), this aspect has received little attention





in organic geochemistry, yet is one of the main objectives within the ecological
approach.
**4 Ecologists focus on the dynamics of organisms using organic matter**
Benthic communities are usually partitioned into different size classes
(e.g. macrofauna, meiofauna, microbes; Gage and Tyler, 1992; Gray and Elliot,
2009; Herman et al., 1999), which are often studied by different research
communities having distinct objectives, approaches and tools. Organic matter
delivered to the seafloor fuels benthic food webs: i.e. it represents food for the
animals and the energy substrate for heterotrophic microbes. Microbial
ecologists study the growth of microbes on delivered organic matter (e.g.
bacterial production) and subsequent microbial loss processes, including
predation and viral lysis (Kemp, 1988; 1990; Danovaro et al., 2008, 2011, 2016).
Microbial ecologists also study in detail the identities and activities or organisms
involved in (an)aerobic respiration pathways and the re-oxidation of reduced
metabolites produced during anaerobic organic matter degradation (Canfield et
al., 2005).  Animal ecologists focus on the response of fauna to food delivery, the
diet and growth of animals and transfer of carbon up the food chain to top
consumers (Krumins et al., 2013; Fig. 3). Interactions among food-web members
are considered key to understand carbon flows (Pimm et al., 1991; van Oevelen
et al., 2010).
During the last two decades, isotopically labeled phytodetritus addition
experiments have been performed to identify the organisms involved in the
immediate processing of organic matter delivered to the seafloor (Middelburg,
2014). These studies often covered all size classes (animals and microbes) and
could show that respiration was the major fate and that all size classes directly
profited from recently deposited organic matter (Moodley et al., 2002, 2005a;
Woulds et al., 2007, 2009, 2016; Witte et al., 2003). In other words,
heterotrophic microbes and small and big animals compete for the same food.
The relative share of organisms in the processing of organic matter was in some
systems and for some consumers proportional to the biomass of the benthic size
class, but not always (Moodley et al., 2005a; Woulds et al, 2009, 2016). For
instance, foraminifera, amoebid protozoa, sometimes contribute
disproportionally to short-term carbon processing reflecting high turn-over of an
active community (Moodley et al., 2002; Woulds et al., 2007). The spatial
distribution of resources is also a key factor governing the relative use of
phytodetritus by bacteria vs. animals (van Nugteren et al., 2009b).
**5 Towards a synthesis**
The above discussion on conceptual views within different research disciplines
highlights a few discrepancies and gaps in our knowledge. Secondary production
by animals and microbes is not included in the geochemical view that focuses on
preservation versus mineralization. It is also largely absent from the organic
geochemical literature. Consumption of organic matter is restricted to microbes
in the geochemical view, while the non-fed animals move organic matter,
microbes and particles around by bioturbation and enhance solute transfer by



bio-irrigation activities. The consumption of organic matter eventually results in
compositional changes of the organic matter remaining but there is little
information that the identity of the organism processing the organic carbon
matters much. Whole ecosystem labeling experiments revealed direct flow from
detritus to most benthic consumers and to the dissolved inorganic carbon pool,
but these short-term experimental results cannot directly be compared to the
long-term natural processing of deposited organic matter because long-term
transfers in the food web and eventual carbon preservation cannot be resolved
experimentally.

**5.1 On the consistency of food-web carbon processing and the geochemical burial-respiration partitioning**

Food webs describe the exchange of matter (e.g. carbon or energy) among
different compartments (organisms) within an ecosystem (Pimm et al., 1991; de
Ruiter et al., 1995) and thus formalize the ecological view on carbon processing
(Cole et al., 2006; van Oevelen et al, 2010). The emphasis is on interaction among
organisms and respiration losses are normally lumped into a single carbon
dioxide loss term (Fig. 3 left). Experimental studies using $^{13}C$ labelled
phytodetritus as a tracer of sediment carbon processing showed that animals
and microbes both can assimilate labile carbon directly and confirmed that
respiration is the largest sink (Moodley et al, 2005a; Buhring et al., 2006;
Andersson et al. 2008; Woulds et al. 2009, 2016).  The geochemical budgeting
approach basically distinguishes only between (refractory) carbon preserved
and buried versus labile organic carbon that is respired to carbon dioxide (Glud,
2008; Aller, 2013; Fig. 3 right). These ecological and geochemical concepts are
consistent when the typical timescale of interest is considered. On the time scale
of days to month deposited carbon is processed by the benthic organisms, a
small part is assimilated and the majority is respired. On longer time scales and
when considering steady-state conditions, i.e. constant faunal and microbial
biomass, there is no net transfer from the detritus pool to the living biomass pool
and all labile carbon is eventually respired because organisms only represent a
(temporary) carbon sink when their biomass is building up or when remains of
secondary producers are buried.

**5.2 Secondary production and the formation of molecularly uncharacterizable organic matter.**

The mere presence of living organisms in sediments clearly indicates that
secondary production is omnipresent. Microbes usually dominate living biomass,
but not always, and living biomass typically contributes a few % at most to the
standing stock of total organic carbon in sediments (Herman et al., 1999).
Various types of experimental evidence have shown that carbon flow through
living compartment is much higher than through the non-living sediment organic
matter pool. Short-term, in situ experiments using $^{13}C$ and/or $^{15}N$ labelled
organic matter (e.g. phytodetritus) revealed rapid incorporation of $^{13}C/^{15}N$ in
physically separated organisms (macro- and meiofauna and foraminifera) and
microbes, the latter via incorporation of tracers in biomarkers specific for certain
microbial groups (Middelburg et al., 2000; Boschker and Middelburg, 2002;

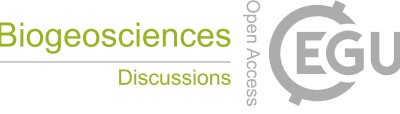

Veuger et al., 2007; Oakes et al., 2012;, Woulds et al., 2007, 2016) Similarly,
ammonium isotope dilution studies have shown that net ammonification is only
a small fraction of the total ammonium regeneration because most of the
ammonium liberated is re-assimilated by the microbial community (Blackburn
and Henriksen, 1983).  Clearly the microbes and animals living in sediment
assimilate carbon and synthesize new tissues. How can this be reconciled with
the geochemical and organic geochemical views in which organic matter is either
preferentially degraded to carbon dioxide or selective preserved (Fig. 1, 3).
These two apparently inconsistent views are consistent if all of the newly
produced organic matter is eventually degraded.
Detailed investigations of organic matter composition might in principle
resolve this issue as microbial and animal processing of organic matter results in
the formation of distinct compounds (Bradshaw et al., 1990; Sun et al., 1999;
Thomas and Blair, 2002; Woulds et al., 2012, 2014). There are few issues with
this approach: (1) most sedimentary organic matter is molecularly
uncharacterizable and the origin (imported from the water column vs. newly
produced within the sediment) can thus not be investigated, (2) microbes living
within (the guts) of animals may mask the animal signatures (Woulds et al.,
2012, 2014) and (3) different analytical windows (amino acids vs. lipids) may
result in different inferences. On the one hand, the accumulation of bacterial
derived non-protein amino acids and peptidoglycan derived D amino acids are
clear signs that extensively modified organic matter contains a major fraction
that is derived from (heterotophic) bacteria (Cowie and Hedges, 1994; Dauwe et
al., 1999; Grutters et al., 2001; Lomstein et al., 2006; Keil and Fogel, 2001; Keil et
al., 2000). On the other hand, using a combined lipid-isotope approach Hartgers
et al. (1994) reported only a minor contribution of bacteria to sedimentary
organic carbon pools. However, Gong and Hollander (1997) used fatty acids and
identified a substantial microbial contribution to sedimentary organic matter.
Secondary production has potentially major consequences for the
interpretation of sedimentary records. If microbial reworking of deposited
organic matter represents a major carbon processing flow and part of the
material is preserved then one would expect that bulk organic matter properties
such as C, N, P elemental ratios and nitrogen isotopes would reflect this.
Degradation of organic matter usually results in the preferential release of
nitrogen and phosphor relative to carbon. Microbes normally have lower C/N
ratios than their substrate, implying that secondary production and
accumulation of microbial derived organic matter results in a net decrease of
sediment C/N ratios (Müller, 1977). In contrast, the C/P ratio of heterotrophic
microbes is rather variable because P demands depends on the growth rate
(Sterner and Elser, 2002) and slowly growing benthic microbes may have high
C:P ratios (Steenbergh et al., 2013). Moreover, microbial P storage also depends
on redox conditions with the consequences that sedimentary C:P ratios are
highly variable (Algeo and Ingall, 2007). Sediment $\delta^{15}N$ values often show a post-
depositional shift towards heavier values in alternating oxic/anoxic settings
(Moodley et al., 2005b). Such a shift is to be expected because regenerated
ammonium is either transformed into nitrite/nitrate (nitrification) or re-
assimilated by the microbial community. During oxic conditions nitrification
occurs with preference for $^{14}N$ and the remaining ammonium available for re-
assimilation by microbes will be relatively rich in $^{15}N$, while during anoxic



conditions oxidation of ammonium is less important or absent, and the
ammonium re-assimilated will have similar $\delta^{15}N$ values as that regenerated.
Secondary production of microbial biomarkers within sediment may also impact
the interpretation of paleorecords (Schouten et al., 2010).
To reconcile the strong experimental evidence for preferential
degradation (Middelburg, 1989), selective preservation (Tegelaar et al., 1989)
and formation of new compounds by secondary producers (Lomstein et al, 2012;
Braun et al, 2017) we present a new integral concept (Fig. 4). Phytodetritus
delivered to sediments is preferentially degraded with the result that new
biomass is formed and that some compounds are selectively preserved. The
newly formed biomass is after death of the organism added to the pool of
degraded detritus and subject to further microbial processing. After multiple
cycles of processing by benthic heterotrophs most of the remaining organic
matter becomes molecularly uncharacterizable. This conceptual model is
consistent with the ruling paradigms of preferential degradation and selective
preservation as well as with the occurrence of secondary production and
formation of molecular uncharacterizable organic matter, but the next step is to
quantify this conceptual view. One approach would be to use proxies for organic
matter degradation state such as fraction of total nitrogen present as amino acid,
non-protein amino acids accumulation and the degradation index (Cowie and
Hedges, 1994; Dauwe and Middelburg, 1998; Dauwe et al., 1999). Lomstein et al.
(2012) and Braun et al. (2017) used amino acid racemization to quantify
turnover of living microbial biomass as well as of bacterially derived organic
matter (necromass) in the deep biosphere.  Veuger et al. (2012) executed a
$^{13}C/^{15}N$ tracer experiment and followed the isotope labels into carbohydrates,
amino acids and lipids and basically showed that most of the deliberately added
heavy isotopes were recovered from the molecularly uncharacterizable pool
within a few weeks and remained in that pool till the end of the experiment (>
1year). Their study provided direct evidence for rapid formation of new
microbial biomass and subsequent transfer of microbial biomass to the pool of
molecular uncharacterizable organic matter. Moreover, the efficient retention of
label was indicative of recycling of molecules (or parts thereof) by microbes
rather than de novo synthesis, consistent with findings in soil science (Dippold
and Kuzyakov, 2016).
**5.3 Animals and carbon supply to sediments**
Marine sediments are often considered donor-controlled systems, i.e. organic
matter is delivered via settling of organic matter produced in the sunlit upper
part of the ocean (Fig. 3) and the consuming sediment communities have no
control on its carbon delivery (Fig. 5). It is only on the time-scale of ocean
bottom-water renewal (10-100 years) that nutrient regenerated by benthic
organisms may impact primary producer in the sunlit upper part of the ocean
(Soetaert et al., 2000). This is obviously different for sediments in the photic
zone that make up about 1/3 of the coastal ocean (Gattuso et al, 1996) because
animals can directly graze and consume the benthic primary producers at the
sediment surface (Middelburg et al., 2000; Evrard et al., 2010, 2012; Oakes et al.,



2012; Fig. 5). Donor and consumer controlled food webs have intrinsic different
dynamics.
Animals living in sediments below the photic zone can in multiple ways
impact carbon processing within marine sediments (Fig. 5). Deposit-feeding
animals mix particles (and thus particulate organic carbon) as a consequence of
their activities. In the case of constant organic carbon delivery (donor-control)
bioturbation stimulates organic carbon processing at depth (Herman et al.,
1999).  In coastal systems, organic matter delivery is more complex because of
multiple deposition-resuspension events and lateral transport pathways. Rice
and Rhoads (1989) showed that in this case (with constant organic carbon
concentration in the top layer) more bioturbation will increase the organic
matter flux into the sediment. Moreover, organic carbon gradients with depth
are steeper for high-quality than low-quality material and bioturbation thus
results in transfer of high-quality organic matter to (micro-)organisms living at
depth.  Animals living in permeable sediment can via surface sediment
topography (bioroughness) induce pore-water flows resulting in the trapping of
phytoplankton (Huettel et al., 2014).
Tropical and cold-water corals, coastal and deep-sea sponges, suspension
feeding bivalves and other marine forests communities utilize particulate
organic matter suspended in the water (Herman et al., 1999; Roberts et al., 2006;
Rossi et al., 2017). This organic carbon is used for maintenance respiration and
growth, but part is excreted as faeces or pseudofaeces and becomes then
available for consumers in the sediments. This can result in local hotspots of
biodiversity and microbial activity in the sediments (Herman et al., 1999;
Gutierrez et al., 2003; Cathalot et al., 2015). Moreover, the physical structures
build by these ecosystem engineers impact hydrodynamics with consequences
for local and distant carbon deposition rates. Soetaert et al. (2016) reported
elevated carbon deposition to ocean margin sediments due to cold-water corals
reefs at very large distances.
Some sponges have the capability to take up dissolved organic carbon and
transform it into sponge tissue (de Goeij et al., 2013; Fiore et al., 2017; Hoer et
al., 2017). This sponge tissue and in particular its detritus can be consumed by
benthic organisms. This sponge loop (de Goeij et al., 2013; Rix et al., 2016) is
another example how animals can manipulate the transfer of organic carbon
from the water-column to the sediments (Fig. 5).
**5.4 Animal stimulation of microbes: An inverted microbial loop?**
The microbial loop is a key concept in ocean biogeochemistry (Pomeroy,
1974; Azam et al., 1983). Dissolved organic matter released by phytoplankton,
zooplankton or viral lysis of bacteria, archaea and algae is consumed by
heterotrophic microbes. These heterotrophs are in turn consumed by flagellates,
ciliates and other small consumers that are predated upon by zooplankton
(Azam et al., 1983; Jumars et al. 1989). Energy shunted into the large,
heterogeneous dissolved organic matter pool is in this way made available again
for animals (Fig. 6).
After discovery of this loop in the surface ocean water, research has been
executed to identify and quantify it in sediments (Kemp, 1988, 1990). This



required substantial investment in developing new methods: these studies
basically revealed that predation on sedimentary bacteria was not that
important (Kemp, 1990; Hondeveld et al., 1992; Hamels et al., 2001; Guilini et al.,
2009). Van Oevelen et al., (2006) made a detailed study on the fate of bacterial
production using in situ ¹³C labelling of bacteria. They observed that 8% was lost
by physical processes, 27% was consumed by animal predation, while bacterial
mortality accounted for 65%.   Viruses are the most important loss term for
sedimentary microbes (Danovaro et al, 2009, 2011, 2016) and the viral lysis
products (dissolved organic matter) are consumed by microbes. This results in a
dissolved organic carbon-microbes cycle (Fig. 6). This benthic microbial cycle
represents a dead end in terms of food web topology, because there is little
transfer to higher trophic levels and most carbon is eventually respired as
needed for mass-balance closure on the long term (Fig. 3).
In fact, the benthic microbial cycle represents more an *inverted microbial*
*loop*: rather that animals profit from the microbial loop sensu Azam et al. (1983),
it appears that benthic microbes profit from animals mixing labile organic matter
downwards into the sediments (Fig. 6). Labile organic matter delivered to the
sediment surface is mixed by animals inhabiting the sediments (Fig. 5). The
transfer of high quality organic matter to deeper sediment layer may prime
sediment microbial communities and in this way stimulate degradation of
indigenous organic matter (van Nugteren et al, 2009a, Bianchi, 2011; Hannadis &
Aller, 2017). This inverted microbial loop is a prime example how animals as
ecosystem engineers impact sediment carbon processing (Meysman et al., 2006).
**5.5 Imagine a world without animals.**
An ocean floor inhabited solely by microbes and without animals was likely the
reference state during the first four billion of years of Earth's history (Canfield,
2014; Lenton and Watson, 2011). Moreover, in modern systems with anoxic
bottom waters benthic animals are absent (Rhoads and Morse, 1971; Diaz and
Rosenberg, 1995, 2008; Levin, 2003; Levin et al, 2009). In these systems, organic
matter degradation pathways are different not only because of a lack of oxygen
and use of alternative electron acceptors, but also because bio-irrigation and
bioturbation are absent (Aller and Aller, 1998; Levin et al., 2009; Middelburg and
Levin, 2009). Consequently, microbe-fauna interactions (enhanced carbon
delivery, Fig. 5; animal stimulation of microbes, Fig. 6) are impeded.
This likely is the reason why more organic carbon is buried in sediments
underlying anoxic bottom waters (Hartnett et al., 1998; Hartnett and Devol,
2003; Middelburg and Levin, 2009; Jessen et al., 2017). Moreover, the organic
matter buried in hypoxic and anoxic settings is usually less degraded (Cowie,
2005, Cowie et al., 2009; Vandewiele et al., 2009; Koho et al., 2013; Jessen et al.,
2017). This presence of animals and all their interactions with organic matter
and microbes has consequences for organic carbon processing in marine
sediments and thus the global carbon cycle. It is obvious for any terrestrial
microbiologist that a world with trees and other macrophytes would be different
than one without. Moreover, biological oceanographers and limnologists agree
that zooplankton and other metazoan consumers contribute to biogeochemical
cycles (Vanni, 2002; Vanni and McIntyre, 2016), and I hope that colleagues



studying marine sediments are aware that BIO in sediment biogeochemistry is
more than just microbiology.

*Acknowledgement*
This paper presents my acceptance lecture for the Vladimir Ivanovich Vernadsky
Medal 2017 of the European Geosciences Union. I thank the colleagues that
nominated me for this award and the many colleagues, students and
postdoctoral fellows with which I have had the pleasure to interact and
collaborate over the years. In particular, the late Carlo Heip who was
instrumental for getting the BIO into my biogeochemistry and my long-term
collaborators in modelling benthic ecosystems (Karline Soetaert, Peter Herman,
Filip Meysman and Bernie Boudreau), in food-web studies (Dick van Oevelen,
Leon Moodley), in organic geochemistry (Jaap Sinnighe Damsté and Stefan
Schouten) and in coastal biogeochemistry (Jean-Pierre Gattuso and Carlos
Duarte). Ton Marcus is thanked for graphic support. This is a contribution to the
Netherlands Earth System Science Centre supported by the Dutch ministry of
Education and Science.





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



Figure 1. Different views, approaches and interests on carbon processing in marine
sediments. Paleoceanographers focus on the sedimentary record,
biogeochemists quantify carbon burial and recycling, organic geochemists study
alteration of organic matter and ecologists focus on carbon as food for organisms
living in the sediment
Figure 2.  Conceptual model of organic matter (OM) degradation and re-oxidation
pathways (based on Middelburg and Levin, 2009).
Figure 3. Carbon processing in marine sediments on the short-term (left) and the long-
term (right). Organic matter produced in the sunlit layer of the ocean and
delivered to the sediments is either consumed by organisms or buried. The
organic matter consumed by organisms is used to synthesize biomass or
metabolized to carbon dioxide and nutrients. On the long-term or at steady-
state, i.e. the biomass of benthic organisms does not change, the benthic
community can be considered a black box diverting organic matter either into
metabolites or into the geosphere (burial).
Figure 4. Conceptual diagram showing the relationships between molecular
uncharacterizable organics, deposited phytodetritus and  secondary production.
Phytodetritus is degraded preferentially and new biomass is formed, which after
death of the organisms is added to the pool of detritus and subject to
degradation. Multiple cycles of organic matter processing eventually results in
the formation of molecular uncharacterizable organic matter.
Figure 5. Organic matter supply to sediments.  1. The traditional view of organic matter
settling passively from the water column (donor control). 2. Sediments in the
photic zone are inhabited by benthic microalgae that produce new organic
matter in situ and grazing animals can impact the growth of these organisms. 3.
Bioturbating animals transfer labile carbon from the sediment surface layer to
deeper layers in the sediments. 4. Suspension feeding organisms enhance the
transfer of suspended particulate matter from the water column to the
sediments (biodeposition). 5. Sponge consume dissolved organic carbon and
produce cellular debris that can be consumed by benthic organisms (i.e. the
sponge loop).
Figure 6. The microbial and inverted microbial loop. In the water column dissolved
organic carbon derived from phytoplankton, zooplankton or microbes (via viral
loop) is consumed by heterotrophic microbes, which in turn are consumed by
protists and small animals with the consequence that carbon flowing through
dissolved organic carbon pools eventually can be used by larger animals
(microbial loop).  In sediments, the dissolved organic carbon (from viral lysis
and other sources) is also consumed by heterotrophic microbes but this carbon
is inefficiently transferred to animals. The engineering activities of animals are
key in delivering labile organic matter (phytodetritus) to microbes living in the
subsurface (inverted microbial loop).





Figure 1. Different views, approaches and interests on carbon processing in marine
sediments. Paleoceanographers focus on the sedimentary record,
biogeochemists quantify carbon burial and recycling, organic geochemists study
alteration of organic matter and ecologists focus on carbon as food for organisms
living in the sediment

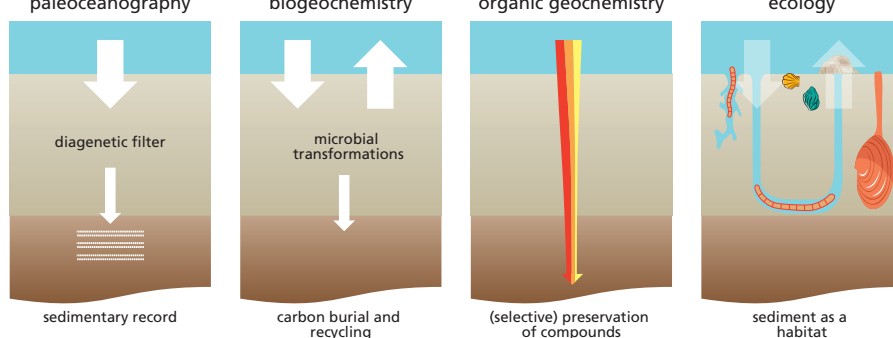




Figure 2. Conceptual model of organic matter (OM) degradation and re-oxidation
pathways (based on Middelburg and Levin, 2009).

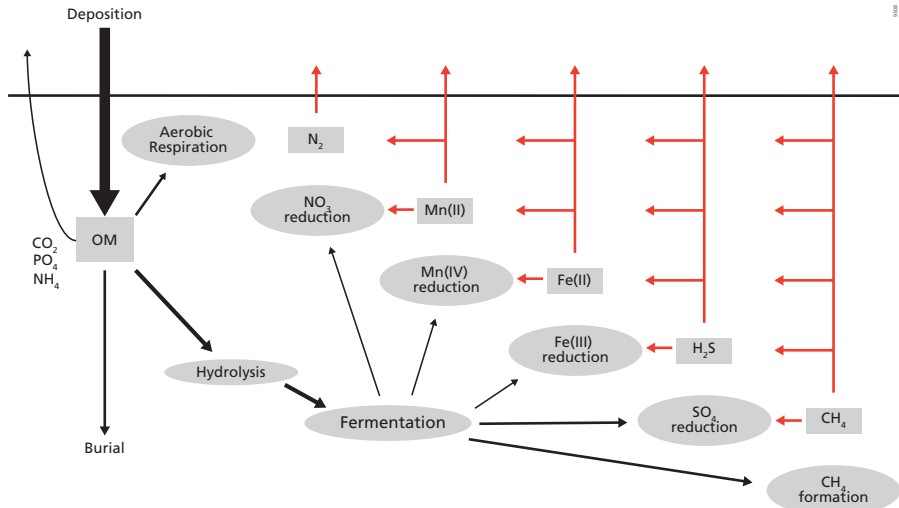






Figure 3. Carbon processing in marine sediments on the short-term (left) and the long-
term (right). Organic matter produced in the sunlit layer of the ocean and
delivered to the sediments is either consumed by organisms or buried. The
organic matter consumed by organisms is used to synthesize biomass or
metabolized to carbon dioxide and nutrients. On the long-term or at steady-
state, i.e. the biomass of benthic organisms does not change, the benthic
community can be considered a black box diverting organic matter either into
metabolites or into the geosphere (burial).

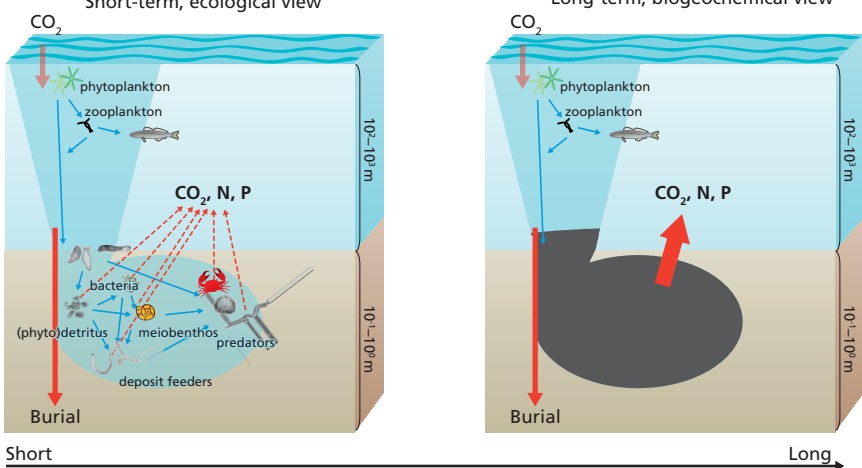




Figure 4. Conceptual diagram showing the relationships between molecular
uncharacterizable organics, deposited phytodetritus and  secondary production.
Phytodetritus is degraded preferentially and new biomass is formed, which after
death of the organisms is added to the pool of detritus and subject to
degradation. Multiple cycles of organic matter processing eventually result in the
formation of molecular uncharacterizable organic matter.

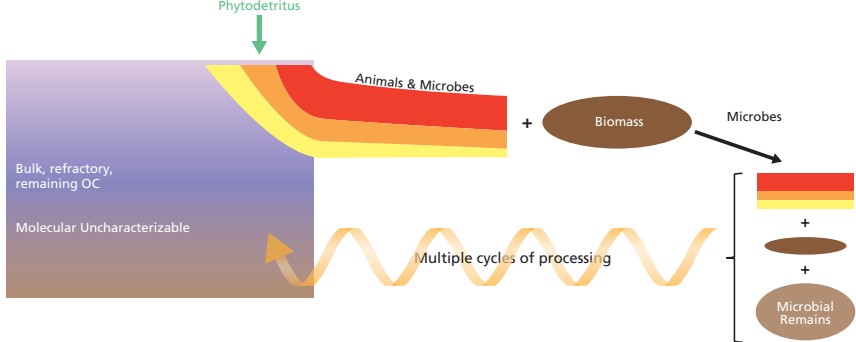





Figure 5. Organic matter supply to sediments. 1. The traditional view of organic matter
settling passively from the water column (donor control). 2. Sediments in the
photic zone are inhabited by benthic microalgae that produce new organic
matter in situ and grazing animals can impact the growth of these organisms. 3.
Bioturbating animals transfer labile carbon from the sediment surface layer to
deeper layers in the sediments. 4. Suspension feeding organisms enhance the
transfer of suspended particulate matter from the water column to the
sediments (biodeposition). 5. Sponge consume dissolved organic carbon and
produce cellular debris that can be consumed by benthic organisms (i.e. the
sponge loop).

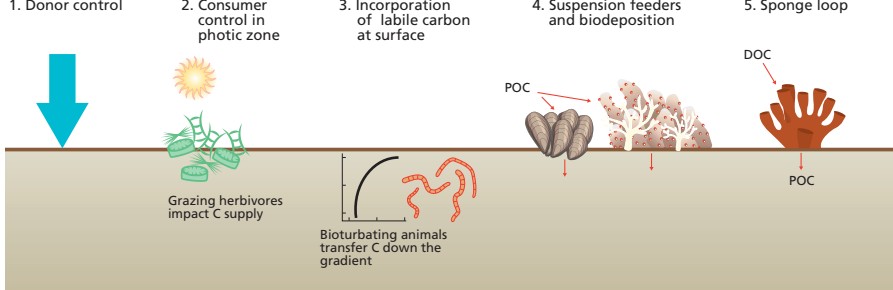






Figure 6. The microbial and inverted microbial loop. In the water column dissolved
organic carbon derived from phytoplankton, zooplankton or microbes (via viral
loop) is consumed by heterotrophic microbes, which in turn are consumed by
protists and small animals with the consequence that carbon flowing through
dissolved organic carbon pools eventually can be used by larger animals
(microbial loop).  In sediments, the dissolved organic carbon (from viral lysis
and other sources) is also consumed by heterotrophic microbes but this carbon
is inefficiently transferred to animals. The engineering activities of animals are
key in delivering labile organic matter (phytodetritus) to microbes living in the
subsurface (inverted microbial loop).

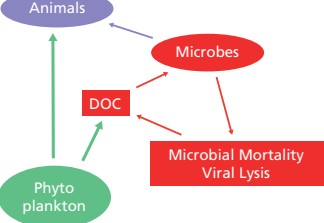

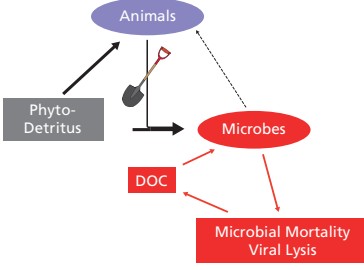
