# Peer review of "Reviews and Synthesis: To the bottom of carbon processing at the seafloor Jack J. Middelburg"

_Biogeosciences, 2017_

## Referee Comment (RC1) · Anonymous Referee #1 · 15 Sep 2017

This paper describes well the linkages between biogeochemistry, organic geochemistry and ecology at the seafloor. It is obvious that these disciplines have strengths and weaknesses, and if they are combined, a much more detailed view of the variety of benthic processes and their interactions can be obtained. The paper emphasizes this in an excellent manner.

Major concerns

My major concern deals with the inclusion of paleoceanography among the disciplines under study. Actually, this discipline is only mentioned in the abstract and the beginning of the introduction, while there is no mention of this in the remaining part of the manuscript. I recommend omitting paleoceanography from the abstract, introduction and figure 1. The focus should then be on biogeochemistry, organic geochemistry and

ecology throughout the paper.

It seems implicit from the text that the research in the three disciplines has been done in isolation only (see lines 64-72). The author should mention that many papers actually have focused on the interface of two or even all three disciplines.

In chapter 2, the title starts with "Geochemists focus on. . .". In this chapter and throughout the manuscript the term geochemistry is used. I suppose that it should be biogeochemistry? Otherwise it makes no sense with the introduction of the paper. It becomes even more tricky when the next chapter deals with organic geochemistry. The reader can easily be confused. I recommend using "biogeochemistry" and "organic geochemistry" to separate these disciplines throughout as defined in the introduction and figure 1.

The introduction of bioturbation in lines 116-122 is not fully clear and does not follow the recommendation and definition introduced by Kristensen et al. (MEPS 446: 285-302, 2012). In this paper, it was argued that bioturbation covers both particle reworking and bioirrigation driven by ventilation. I recommend following this definition throughout the paper. It does not change the meaning, but merely clarifies the terms. Moreover, it is striking that only Aller is cited when ventilation is discussed. Other and more recent papers have been published on this subject.

I miss the role of ventilation driven bioirrigation in chapter 5.4 where animal stimulation of microbes is dealt with. Several papers have shown that the subduction of oxygen into burrows by animal ventilation has the capacity to enhance microbial processes, including a stimulation of decomposition of old and buried organic matter, by up to one order of magnitude. I recommend that this aspect is dealt with here.

Minor points

Line 19: Change to "It is shown. . ." Line 51-52: Change to ". . .of substances that have survived. . ." Line 54: Change to ". . .of the material that is eventually. . ." Line 55-

57: Change to "...remaining part is recycled as key nutrients (e.g. N, P, Si, Fe) to sustain primary production in the water column..." Line 64: Change to "...topic, e.g. fate of organic matter..." Line 69: I disagree with this definition of bioturbation – see above. Line 72: Delete "their food" Line 77: Now we are introduced to four disciplines (two new!!): Organic geochemistry, sediment geochemistry, ecology and microbiology!! Please place them into the context of the three primary disciplines the entire story deals with. It is particularly important as this statement is presented in the last paragraph of the introduction. Line 86-87: Change to "...overlying water, or is buried..." Line 92: Change to "...rates are high in vegetated..." Line 96-97: Change to "...degraded to inorganic carbon at rates that provide an..." Line 130: Now we are introduced to microbial ecology as a discipline – how is that connected to the three major disciplines of this paper? Line 134-138: These lines make no real sense and should be rephrased or omitted. Line 139-140: Change to "...and porewater depth profiles..." Line 149: Change to "...and anaerobic organisms, including the distribution..." Line 152: Delete "these" Line 177-178: Change to "...or its reverse, the refractory nature of organic matter, with various..." Line 179: Change to "...amino acids..." Line 186-187: Change to "...organic matter by bacteria (Cowie and Hedges...." Line 190-191: Change to "...importance of bacteria and fauna for organic matter..." Line 193-194: Change to "...secondary production (Cowie and Hedges..." Line 202: Change to "...(e.g. macrofauna, meiofauna and microbes..." Line 216: Change to "...considered the key to..." Line 218-223: The element dealt with here must be carbon as respiration is mentioned. Please specify. Line 231-233: This last sentence appears to be out of context. Please omit or rephrase. Line 244-245: Change to "...particles around by reworking and enhance solute transfer by ventilation driven bioirrigation." Line 245-248: Change to "The ingestion of food eventually results in compositional changes of the organic matter, but there is little information that the identity of the processing organism matters much." Line 252: Change to "...transfers within the food web..." Line 287-288: change to "...flow through the living compartment is much higher..." Line 292-293: Change to "....and microbes (Middelburg et al..." Line 313: Change to "On one hand..." Line

318-321: These statements are quite contradictory. Which one to believe? Line 323-324: Change to "...microbial processing of deposited organic matter represents a major carbon flow and..." Line 328: Change to "phosphorus" Line 327-331: Somehow these lines are in conflict. First it is mentioned that decomposition results in preferential release of N and P, then it is implied that accumulation of microbial organic matter decreases the C/N ratio. These two processes cannot occur simultaneously. Please clarify. Line 328-335: Please be consistent in the notation. In this section, ratios are partly notated C/N and C/P or C:N and C:P. Use the same throughout. Line 350: Who are "we"? The paper has only one author! Line 386: Change to "...primary producers..." Line 401: Delete "more" Line 420-425: Just a comment – It has long been known that all invertebrates have the capacity to take up DOC. So, this is not only valid for sponges. Line 473: Change to "...also because bioturbation is absent..." Figure 1: Please omit the paloceanography panel – it is not dealt with in the text. Figure 4: Please explain what the red-orange-yellow colors stand for.

---

## Referee Comment (RC2) · R. Danovaro (Referee) · 19 Sep 2017

This paper is an invited contribution by Prof. Jack J. Middelburg, presenting his acceptance lecture for the Vladimir Ivanovich Vernadsky Medal 2017 of the European Geosciences Union. The paper addresses timely and crucially important scientific questions, reconciling different views and providing an unifying vision to the biogeochemical processes involving C cycling at the seafloor. This is a seminal contribution, introducing novel concepts, opening new perspectives in different scientific fields from geology to biogeochemistry, organic chemistry and ecology. The figures are excellent to clarify complex concept and very useful for teaching. This paper is therefore a fantastic contribution to the the knowledge and reflections on the carbon processing at the seafloor. I have only few and minor suggestion to be eventually considered for the preparation

of the final version: P1 L19-22. "It shown [. . .] time scales." It refers to previously published papers or reflects the personal opinion of the author? P1 L22. "It is hypothesized..". It has been hypothesized by other authors? P1 L27. ". . .is presented. . ." Similarly, I feel that using the first person here would sound better. P1 L47. Is only carbon processing involved? P2 L64-72. Is there any reference available in the literature. P2 L84. pls check "eventually is either mineralized into metabolites (carbon dioxide and nutrients)... P2 L90. closely "and positively"? P3 L116-119. Please check the sentence P3 L128. Please delete "is" P4 L150-152. "Chemoautotrophs in sediments [. . .] to the ocean (Middelburg, 2011)". This sentence sounds a bit isolated from the rest of the text, could you add a few words to better link it to the rest of the discussion. P4 L174. "the degraded fraction cannot be studied" could be "the degraded fraction cannot be easily studied" P4 L177-178. "its reverse the" can be deleted. P4 L191. Please avoid repeating "pathways". P4 L194. "microbes and animals and other heterotrophs". Better say "heterotrophic organisms"? P5 L196. ". . .yet is. . .". "it" or "this" is missing in between? P5 L218. The papers by Nomaki et al. (e.g., MEPS, 2006, "Different ingestion patterns of 13C-labeled bacteria and algae by deep-sea benthic foraminifera.") and by Sweetman and colleagues (e.g., Sweetman & Witte, MEPS, 2008, "Response of an abyssal macrofaunal community to a phytodetrital pulse") could be cited in this paragraph. P5 L199-233. Any suggestion on how to make ecological studies more complete, better harmonized with the other approaches? P6 L285-286. "Microbes usually dominate living biomass but not always". Here you could expand a bit the concept by referring to the increasing relevance of microbial biomass with increasing water depth, referring to Rex et al., MEPS, 2006, "Global bathymetric patterns of standing stock and body size in the deep-sea benthos"). This aspect has been highlighted also in Danovaro et al., TREE, 2014, "Challenging the paradigms of deep-sea ecology". The contribution of the biomass of bacteria, archaea and viruses in surface sediments has been recently summarised in Danovaro et al, AME, 2015, "Towards a better quantitative assessment of the relevance of deep-sea viruses, Bacteria and Archaea in the functioning of the ocean seafloor") in case it can be useful for the discussion. P7 L299. Here "Tissue"

could be replaced by "Biomass" ? P7 L301. Selectively "preserved" could be "recycled to bypass de novo synthesis"? or something similar? P7 L327-28. "Degradation of [...] relative to carbon." Can add a citation here? P7 L328. Phosphorus? P8 L350. (and, e.,g., P1 L18) Instead saying "we present a new.." you could say "here it is presented a new..." if you want to avoid the use of the first person. P9 L416. Build could be built? Figure 1 and 4. Do the red/orange/yellow colours refer to specific components/processes? Figure 2. Please add legend of the colour used (if useful). Figure 5. Please explain the axes in the central plot.

---

## Referee Comment (RC3) · D. E. Canfield (Referee) · 27 Oct 2017

Review of Middelburg " Reviews and Synthesis: To the bottom of carbon processing at the seafloor"

This MS is a summary of Middelburg's acceptance speech for the Vladimer Vernadsky prize. I view this as a thoughtful contribution attempting to integrate various views of the processing of organic matter at the seafloor. The approach is commendable and I believe that this will be a valuable framework for scientists, especially young ones, to access and integrate various views on sediment carbon processing. I must admit that nothing in this piece surprised me, and that Middelburg's view on the relationship between life and carbon diagenesis pretty much conforms with the way I have rationalized

these interactions myself. But, never mind, I think many will find this very inspiring.

I have made numerous comments on the .pdf of the MS. Some are stylistic, but many of are more substance. For example, researchers of sediment diagenesis are not actively ignoring the various sub-disciplines that study carbon turnover in sediments, and I think also that many incorporate a wider set of sub-disciplines than implied here. Indeed, one could write a very different paper highlighting the relatively few multi-disciplinary approaches attempting to bridge the disciplinary gaps highlighted here. Many of these papers would be by the author, his students and close associates. I also realize that a paper like this could have 100's of references, but I also think the referencing is a bit thin in places. These points have been highlighted directly on the MS pages. Overall and enjoyable read that should be of great value to the community.

Don Canfield

Please also note the supplement to this comment:
https://www.biogeosciences-discuss.net/bg-2017-362/bg-2017-362-RC3-supplement.pdf

―――――――――――――――

**Supplement:**

[revised manuscript text omitted]

---

## Author Comment (AC1) · 20 Nov 2017

The comment was uploaded in the form of a supplement:
https://www.biogeosciences-discuss.net/bg-2017-362/bg-2017-362-AC1-supplement.pdf

---

## Author Comment (AC2) · 20 Nov 2017

**2: Roberto Danovaro**

I thank Roberto Danovaro for his evaluation and constructive feedback. I repeat the text of the referee in italics and my response follows below in normal font.

*This paper is an invited contribution by Prof. Jack J. Middelburg, presenting his acceptance lecture for the Vladimir Ivanovich Vernadsky Medal 2017 of the European Geo- sciences Union. The paper addresses timely and crucially important scientific ques-tions, reconciling different views and providing an unifying vision to the biogeochemical processes involving C cycling at the seafloor. This is a seminal contribution, introducing novel concepts, opening new perspectives in different scientific fields from geology to biogeochemistry, organic chemistry and ecology. The figures are excellent to clarify complex concept and very useful for teaching. This paper is therefore a fantastic contribution to the the knowledge and reflections on the carbon processing at the seafloor.*

Thanks for these very kind words.

*I have only few and minor suggestion to be eventually considered for the preparation of the final version:P1 L19-22. "It shown [...] time scales." It refers to previously published papers or reflects the personal opinion of the author?P1 L22. "It is hypoth- esized..". It has been hypothesized by other authors? P1 L27. "...is presented..." Similarly, I feel that using the first person here would sound better. P1 L47. Is only car- bon processing involved? P2 L64-72. Is there any reference available in the literature. P2 L84. pls check "eventually is either mineralized into metabolites (carbon dioxide and nutrients)... P2 L90. closely "and positively"? P3 L116-119. Please check the sentence P3 L128. Please delete "is" P4 L150-152. "Chemoautotrophs in sediments [. . .] to the ocean (Middelburg, 2011)". This sentence sounds a bit isolated from the rest of the text, could you add a few words to better link it to the rest of the discussion. P4 L174. "the degraded fraction cannot be studied" could be "the degraded fraction cannot be easily studied" P4 L177-178. "its reverse the" can be deleted. P4 L191. Please avoid repeat- ing "pathways". P4 L194. "microbes and animals and other heterotrophs". Better say "heterotrophic organisms"? P5 L196. ". . .yet is. . .". "it" or "this" is missing in between? P5 L218. The papers by Nomaki et al. (e.g., MEPS, 2006, "Different ingestion patterns of 13C-labeled bacteria and algae by deep-sea benthic foraminifera.") and by Sweet- man and colleagues (e.g., Sweetman & Witte, MEPS, 2008, "Response of an abyssal macrofaunal community to a phytodetrital pulse") could be cited in this paragraph. P5 L199-233. Any suggestion on how to make ecological studies more complete, better harmonized with the other approaches? P6 L285-286. "Microbes usually dominate living biomass but not always". Here you could expand a bit the concept by referring to the increasing relevance of microbial biomass with increasing water depth, referring to Rex et al., MEPS, 2006, "Global bathymetric patterns of standing stock and body size in the deep-sea benthos"). This aspect has been highlighted also in Danovaro et al., TREE, 2014, "Challenging the paradigms of deep-sea ecology". The contribution of the biomass of bacteria, archaea and viruses in surface sediments has been recently summarised in Danovaro et al, AME, 2015, "Towards a better quantitative assessment of the relevance of deep-sea viruses, Bacteria and Archaea in the functioning of the ocean seafloor") in case it can be useful for the discussion. P7 L299. Here "Tissue" could be replaced by "Biomass"? P7 L301. Selectively "preserved" could be "recycled to bypass de novo synthesis"? or something similar? P7 L327-28. "Degradation of [. . .] relative to carbon." Can add a citation here? P7 L328. Phosphorus? P8 L350. (and, e.,g., P1 L18) Instead saying "we present a new.." you could say "here it is pre- sented a new..." if you want to avoid the use of the first person. P9 L416. Build could be built? Figure 1 and 4. Do the red/orange/yellow colours refer to specific compo- nents/processes? Figure 2. Please add legend of the colour used (if useful). Figure 5. Please explain the axes in the central plot.*

These corrections and suggestions will improve the paper and will therefore be incorporated. Text will be modified were needed, references will be added and figure captions will be modified to indicate the differences in lability between different organic compounds, etc.

---

## Author Comment (AC3) · 20 Nov 2017

**3: Don Canfield**

I thank Don Canfield for his evaluation and constructive feedback. I repeat the text of the referee in italics and my response follows below in normal font.

*This MS is a summary of Middelburg's acceptance speech for the Vladimer Vernadsky prize. I view this as a thoughtful contribution attempting to integrate various views of the processing of organic matter at the seafloor. The approach is commendable and I believe that this will be a valuable framework for scientists, especially young ones, to access and integrate various views on sediment carbon processing. I must admit that nothing in this piece surprised me, and that Middelburg's view on the relationship between life and carbon diagenesis pretty much conforms with the way I have rationalized these interactions myself. But, never mind, I think many will find this very inspiring.*

Thanks for these kind words and the confirmation that our views on this topic are very similar.

*I have made numerous comments on the .pdf of the MS. Some are stylistic, but many of are more substance. For example, researchers of sediment diagenesis are not actively ignoring the various sub-disciplines that study carbon turnover in sediments, and I think also that many incorporate a wider set of sub-disciplines than implied here.*

Thanks for all the detailed comments annotated on the pdf. These are very helpful and I will likely incorporate most of the suggestions.

*Indeed, one could write a very different paper highlighting the relatively few multi-disciplinary approaches attempting to bridge the disciplinary gaps highlighted here. Many of these papers would be by the author, his students and close associates. I also realize that a paper like this could have 100's of references, but I also think the referencing is a bit thin in places. These points have been highlighted directly on the MS pages. Overall and enjoyable read that should be of great value to the community.*

Including all literature on this topic would indeed require 100s of references and that would be a different type of paper: i.e. more complete/comprehensive but less integrated and readable. I will incorporate the suggestions made by this and the other referees and add references where a need was identified.

---

## Author Response (AR1)

Dear Tina:

Thank you very much for handling this paper. I have uploaded a revised version as well as the original one with all the annotations.

As I wrote on the public pages, I basically incorporated all feedback but for two aspects of referee # 1: A: bioturbation/bio-irrigation terminology: all through I state specifically what I I mean (e.g., sediment reworking/mixing instead of bioturbation). B: suggestion to delete paleoceanography from introduction/Figure 1: I have not done so because that community also looks at carbon processing, although I do not discuss it in detail.

Has the paper improved: Yes, it has (thank you referees). I have included the role of bio-irrigation in stimulation organic carbon processing more explicitly (feedback from #1 and Don Canfield), have included about 15 additional references (all the suggested ones, references added because there was a need identified by the referees and a few more), checked for consistent use of terminology (C:N vs C/N; biogeochemistry vs. geochemistry, etc) and clarified where needed.

I hope that the present version is suitable for publication.

With best regards,

Jack Middelburg

[revised manuscript text omitted]

---

## Editor Decision (ED1)

**5 Bacteria and Marine Biogeochemistry**

Bo Barker Jørgensen

Geochemical cycles on Earth follow the basic laws of thermodynamics and proceed towards a state of maximal entropy and the most stable mineral phases. Redox reactions between oxidants such as atmospheric oxygen or manganese oxide and reductants such as ammonium or sulfide may proceed by chemical reaction, but they are most often accelerated by many orders of magnitude through enzymatic catalysis in living organisms. Throughout Earth's history, prokaryotic physiology has evolved towards a versatile use of chemical energy available from this multitude of potential reactions. Biology, thereby, to a large extent regulates the rate at which the elements are cycled in the environment and affects where and in which chemical form the elements accumulate. By coupling very specifically certain reactions through their energy metabolism, the organisms also direct the pathways of transformation and the ways in which the element cycles are coupled. Microorganisms possess an enormous diversity of catalytic capabilities which is still only incompletely explored and which appears to continuously expand as new organisms are discovered. A basic understanding of the microbial world and of bacterial energy metabolism is therefore a prerequisite for a proper interpretation of marine geochemistry - a motivation for this chapter on biogeochemistry.

The role of microorganisms in modern biogeochemical cycles is the result of a long evolutionary history. The earliest fossil evidence of prokaryotic organisms dates back about 3.7 billion years (Schopf and Klein 1992). Only some 1.5 billion years later did the evolution of oxygenic photosynthesis apparently take place and it may have taken another 1.5 billion years before the oxygen level in the atmosphere and ocean rose to the present-day level, thus triggering the rapid evolutionary radiation of metazoans at the end of the Proterozoic era. Through the two billion years

that Earth was inhabited exclusively by prokaryotic microorganisms, the main element cycles and biogeochemical processes known today evolved. The microscopic prokaryotes developed the complex enzymatic machinery required for these processes and are even today much more versatile with respect to basic types of metabolism than plants and animals which developed over the last 600 million years. In spite of their uniformly small size and mostly inconspicuous morphology, the prokaryotes are thus physiologically much more diverse than the metazoans. In the great phylogenetic tree of all living organisms, humans are more closely related to slime molds than the sulfate reducing bacteria are to the methanogenic archaea. The latter two belong to separate domains of prokaryotic organisms, the *bacteria* and the *archaea*, respectively (the term 'prokaryote' is used rather than 'bacteria' when also the archaea are included). Animals and plants, including all the eukaryotic microorganisms, belong to the third domain, *eukarya*.

**5.1 Role of Microorganisms**

**5.1.1 From Geochemistry to Microbiology – and back**

Because of the close coupling between geochemistry and microbiology, progress in one of the fields has often led to progress in the other. Thus, analyses of chemical gradients in the pore water of marine sediments indicate where certain chemical species are formed and where they react with each other. The question is then, is the reaction biologically catalyzed and which microorganisms may be involved?

An example is the distribution of dissolved ferrous iron and nitrate, which in deep sea sediments often show a diffusional interface between the two species (Fig. 5.1A). Based on such gradients, Froelich et al. (1979), Klinkhammer (1980) and others suggested that $Fe^{2+}$ may be readily oxidized by nitrate, presumably catalyzed by bacteria. Marine microbiologists, thus, had the background information to start searching for nitrate reducing bacteria which use ferrous iron as a reductant and energy source. It took, however, nearly two decades before such bacteria were isolated for the first time and could be studied in pure culture (Straub et al. 1996, Benz et al. 1998; Fig. 5.1B). They appear to occur widespread in aquatic sediments but their quantitative importance is still not clear (Straub and Buchholz-Cleven 1998). The bacteria oxidize ferrous iron according to the following stoichiometry:

$$10FeCO_3 + 2NO_3^- + 24H_2O \rightarrow$$
$$10Fe(OH)_3 + N_2 + 10HCO_3^- + 8H^+ \qquad (5.1)$$

Also the observation of a deep diffusional interface between sulfate and methane in marine sediments has led to a long-lasting search by microbiologists for methane oxidizing sulfate reducers. The geochemical data and experiments demonstrate clearly that methane is oxidized to $CO_2$ several meters below the sediment surface, at a depth where no other potential oxidant than sulfate seems to remain (Reeburgh 1969, Iversen and Jørgensen 1985, Alperin and Reeburgh 1985, Chaps. 3, 8 and 14). Attempts in several laboratories to isolate methane oxidizing sulfate reducers have, however, so far not led to confirmed results, and it is still not clear whether such organisms exist or do not exist. Alternatively, it has been suggested that methane oxidation is a reversal of the metabolic pathway of methane formation from $H_2$ and $CO_2$ in methanogenic organisms (Eq. 5.2). Thermodynamic calculations indicate that this would become the exergonic direction of reaction if the $H_2$ partial pressure were extremely low (Hoehler et al. 1994, 1998). The sulfate reducing bacteria are known to be highly efficient $H_2$ scavengers (Eq. 5.3). When methane diffuses into the sulfate zone, the $H_2$ partial pressure may thus be

[Figure]

**Fig. 5.1** A) Pore water gradients of nitrate, dissolved manganese, and iron in sediments from the eastern equatorial Atlantic at 5000 m depth. The gradients indicate that $Fe^{2+}$ is oxidized by $NO_3^-$, whereas $Mn^{2+}$ may be oxidized by $O_2$ (no data). (Data from Froelich et al. 1979; Station 10GC1). B) Anaerobic bacterial oxidation of ferrous to ferric iron with nitrate in an enrichment culture. Filled symbols show results from a growing culture, open symbols shows a control experiment with killed cells (no concentration changes). Symbols show ferric iron (● + ○) and nitrate (■ + □). (Data from Straub et al. 1996).

low enough that methane could be oxidized to $CO_2$:

$$CH_4 + 2H_2O \Leftrightarrow CO_2 + 4H_2 \qquad (5.2)$$

$$4H_2 + SO_4^{2-} + 2H^+ \rightarrow H_2S + 4H_2O \qquad (5.3)$$

There are also many examples of the reverse inspiration, that progress in microbiology has led to a new understanding of geochemistry. One such example was the discovery of a widespread ability among laboratory cultures of sulfate reducing and other anaerobic bacteria to disproportionate inorganic sulfur compounds of intermediate oxidation state (Bak and Cypionka 1987). By such a disproportionation, which can be considered an inorganic fermentation, elemental sulfur or thiosulfate may be simultaneously oxidized to sulfate and reduced to sulfide:

$$4S^0 + 4H_2O \rightarrow 3H_2S + SO_4^{2-} + 2H^+ \qquad (5.4)$$

$$S_2O_3^{2-} + H_2O \rightarrow H_2S + SO_4^{2-} \qquad (5.5)$$

A search for the activity of such bacteria, by the use of radiotracers and sediment incubation experiments, revealed their widespread occurrence in the sea bed and their great significance for the marine sulfur cycle (Jørgensen 1990, Jørgensen and Bak 1991, Thamdrup et al. 1993). Disproportionation reactions also cause a strong fractionation of sulfur isotopes, which has recently led to a novel interpretation of stable sulfur isotope signals in modern sediments and sedimentary rocks with interesting implications for the evolution of oxygen in the global atmosphere and ocean (Canfield and Teske 1996). The working hypothesis is that the large isotopic fractionations of sulfur, which in the geological record started some 600-800 million years ago, are the result of disproportionation reactions. From modern sediments we know that these conditions require two things: bioturbation and an efficient oxidative sulfur cycle. Both point towards a coupled evolution of metazoans and a rise in the global oxygen level towards the end of the Proterozoic and start of the Cambrian.

**5.1.2  Approaches in Marine Biogeochemistry**

The approaches applied in marine biogeochemistry are diverse, as indicated in Figure 5.2, and range from pure geochemistry to experimental ecology, microbiology and molecular biology. Mineral phases and soluble constituents are analyzed and the data used for modeling of the diagenetic reactions and mass balances (Berner 1980, Boudreau 1997, Chap. 14). Dynamic processes are studied in retrieved sediment cores, which are used to analyze solute fluxes across the sediment-water interface or to measure process rates by experimental approaches using, e.g. radiotracers, stable isotopes or inhibitors. Studies are also carried out directly on the sea floor using advanced instrumentation such as autonomous benthic landers, remotely operated vehicles (ROV's), or manned submersibles. Benthic landers have been constructed which can be deployed on the open ocean from a ship, sink freely to the deep sea floor, carry out pre-programmed measurements while storing data or samples, release ballast and ascend again to the sea surface to be finally retrieved by the ship (Tengberg et al. 1995). Such in situ instruments may be equipped with A) samplers and analytical instruments for studying the benthic boundary layer (Thomsen et al. 1996), B) microsensors for high-resolution measurements of chemical gradients in the sediment (Reimers 1987; Gundersen and Jørgensen 1990), C) flux chambers for measurements of the exchange of dissolved species across the sediment-water interface (Smith et al. 1976, Berelson et al. 1987), D) coring devices for tracer injection and measurements of processes down to 0.5 meter sediment depth (Greeff et al. 1998).

Progress in microsensor technology has also stimulated research on the interaction between processes, bacteria and environment at a high spatial resolution (cf. Sect. 3.4). Microsensors currently used in marine research can analyze $O_2$, $CO_2$, pH, $NO_3^-$, $Ca^{2+}$, $S^{2-}$, $H_2S$, $CH_4$ and $N_2O$ as well as physical parameters such as temperature, light, flow or position of the solid-water interface (Kühl and Revsbech 1998). Sensors have mostly been based on electrochemical principles. However, microsensors based on optical fibers (optodes) or sensors based on enzymes or bacteria (biosensors) are gaining importance. Examples of microsensor data from marine sediments are given in Chapter 3.

Among the important contributions of microbiologists to biogeochemistry is to quantify the populations of bacteria, isolate new types of microorganisms from the marine environment, study their physiology and biochemistry in laboratory

cultures, and thereby describe the microbial diversity and metabolic potential of natural microbial communities. There are currently about a thousand species of prokaryotes described from the sea and the number is steadily increasing. The rather time-consuming task of isolating and describing new bacterial species, however, sets a limit to the rate of progress. The classical microbiological approaches also have shortcomings, among others that only the viable or culturable bacteria are recognized, and that the counting procedures for these viable bacteria, furthermore, underestimate the true population size by some orders of magnitude.

In recent years, the rapid developments in molecular biology have opened new possibilities for the study of bacterial populations and their metabolic activity, even at the level of individual cells. A new taxonomy of the prokaryotic organisms has developed at a genetic level, based on sequence analyses of ribosomal RNA, which now allows a phylogenetic identification of microorganisms, even of those which have not yet been isolated and studied in laboratory cultures. This has revealed a much greater species diversity than had been anticipated just a decade ago and has for the first time enabled a true quantification of defined bacterial groups in nature. By the use of molecular probes, which bind specifically to RNA or DNA of selected target organisms, both cultured and uncultured bacteria can now be identified phylogenetically. When such probes are

[Figure]

**Fig. 5.2**  Approaches of marine biogeochemistry. Top: Geochemical methods based on solute and solid phase analyses and modelling. Left: Experimental methods for the analyses of process rates. Bottom: Identification, quantification and characterization of the microbial populations. Right: High resolution and in situ methods for the analyses of microbial populations and their microenvironment. (Graphics by Niels Birger Ramsing and reproduced from Jørgensen 1996).

fluorescently labelled, a sediment sample may be stained by the probes and individual cells of the target bacteria can then be observed and counted directly under a fluorescense microscope. This technique is called Fluorescent In Situ Hybridization (FISH) and is rapidly gaining importance in biogeochemistry.

An example of such a FISH quantification is shown in Figure 5.3 (Llobet-Brossa et al. 1998). The total cell density of microorganisms in a sediment from the German Wadden Sea was up to $4 \cdot 10^9$ cells cm$^{-3}$, which is typical of coastal marine sediments. Of these cells, up to 73% could be identified as eubacteria and up to 45% were categorized to a known group of eubacteria. The sulfate reducing bacteria comprised some 10-15% of the identified microorganisms, while other members of the group proteobacteria comprised 25-30%. Surprisingly, members of the *Cytophaga-Flavobacterium* group were the most abundant in all sediment layers. These bacteria are specialized in the degradation of complex macromolecules and their presence in high numbers indicates their role in the initial hydrolytic degradation of organic matter.

Such studies on the spatial distribution of bacteria provides information on where they may be actively transforming organic or inorganic compounds, and thereby indicate which processes are likely to take place. This is particularly important when chemical species are rapidly recycled so that the dynamics of their production or consumption is not easily revealed by geochemical analyses. The introduction of high resolution tools such as microsensors and molecular probes has helped to overcome one of the classical problems in biogeochemistry, namely to identify the relationship between the processes that biogeochemists analyze, and the microorganisms who carry them out. The magnitude of this problem may be appreciated when comparing the scale of bacteria with that of humans. The bacteria are generally 1-2 μm large, while we are 1-2 m. As careful biogeochemists we may use a sediment sample of only 1 cm$^3$ to study the metabolism of, e.g. methane producing bacteria. For the study of the metabolism of humans, this sample size would by isometric analogy correspond to a soil volume of 1000 km$^3$. Thus, it is not surprising that very sensitive methods, such as the use of radiotracers, are often necessary when bacterial processes are to be demonstrated over short periods of time (hours-days). It is also obvious, that a 1 cm$^3$ sample will include a great diversity of prokaryotic

[Figure]

**Fig. 5.3**  Distribution of microorganisms in sediment from the German Wadden Sea determined by total cell counts with DAPI staining and by fluorescent in situ hybridization of eubacteria. The color zonation of the sediment is indicated. (Data from Llobet-Brossa et al. 1998).

organisms and metabolic reactions and that a much higher resolution is required to sort out the activities of individual cells or clusters of organisms.

**5.2    Life and Environments at Small Scale**

The size spectrum of living organisms and of their environments is so vast that it is difficult to comprehend (Fig. 5.4). The smallest marine bacteria with a size of ca 0.4 µm are at the limit of resolution of the light microscope, whereas the largest whales may grow to 30 m in length, eight orders of magnitude larger. The span in biomass is nearly the third power of this difference, $<(10^8)^3$ or about $10^{22}$ (whales are not spherical), which is comparable to the mass ratio between humans and the entire Earth. It is therefore not surprising that the world as it appears in the microscale of bacteria is also vastly different from the world we humans can perceive and from which we have learnt to appreciate the physical laws of nature. These are the classical laws of Newton, relating mass, force and time with mass

movement and flow and with properties such as acceleration, inertia and gravitation. As we go down in scale and into the microenvironment of marine bacteria, these properties lose their significance. Instead, viscosity becomes the strongest force and molecular diffusion the fastest transport.

**5.2.1    Hydrodynamics of Low Reynolds Numbers**

Water movement on a large scale is characterized by inertial flows and turbulence. These are the dominant mechanisms of transport and mixing, yet they are inefficient in bringing substrates to the microorganisms living in the water. This is because the fundamental property required for turbulence, namely the inertial forces associated with mass, plays no role for very small masses, or at very small scales, relative to the viscosity or internal friction of the fluid. In the microscale, time is insignificant for the movement of water, particles or organisms and only instantaneous forces are important. Furthermore, fluid flow is rather simple and predictable, the water sticking to any solid surface and adjacent water volumes slipping past it in a smooth pattern of laminar

[Figure]

Fig. 5.4    Relationships between size, diffusion time and Reynolds number. Representative organisms of the different scales and their length and biomass are indicated. The diffusion times were calculated for small solutes with a molecular diffusion coefficient of $10^{-5}$ cm$^2$ s$^{-1}$. The Reynolds numbers were calculated for organisms swimming at a speed typical for their size or for water parcels of similar size and moving at similar speed. (From Jørgensen 1996).

flow. The transition from laminar to turbulent flow depends on the scale and the flow velocity and is described by the dimensionless Reynolds number, $Re$, which expresses the ratio between inertial and viscous forces affecting the fluid or the particle considered:

$$Re = uL/v \qquad (5.6)$$

where $u$ is the velocity, $L$ is the characteristic dimension of the water parcel or particle, and $v$ is the kinematic viscosity of the seawater (ca 0.01 cm$^2$ s$^{-1}$ at 20°C). The transition from low (<1) to high Reynolds numbers for swimming organisms, for sinking particles, or for hydrodynamics in general lies in the size range of 0.1-1 mm.

**5.2.2    Diffusion at Small Scale**

Diffusion is a random movement of molecules due to collision with water molecules which leads to a net displacement over time, a 'random walk'. When a large number of molecules is considered, the mean deviation, $L$ (more precisely: the root mean square of deviations), from the starting position is described by a simple but very important equation, which holds the secret of diffusion:

$$L = \sqrt{2Dt} \qquad (5.7)$$

where $D$ is the diffusion coefficient and $t$ is the time. This equation says that the distance molecules are likely to travel by diffusion increases only with the square root of time, not with time itself as in the locomotion of objects and fluids which we generally know from our macroworld. Expressed in a different way, the time needed for diffusion increases with the square of the distance:

$$t = \frac{L^2}{2D} \qquad (5.8)$$

These are very counter-intuitive relations which have some surprising consequences. From Equation 5.7 one can calculate the 'diffusion velocity', which is distance divided by time:

$$\text{'Diffusion velocity'} = L/_t = \sqrt{2Dt} \qquad (5.9)$$

This leads to the curious conclusion that the shorter the period over which we measure diffusion, the larger is its velocity and vice versa. This is critical to keep in mind when working with

Table 5.1   Mean diffusion times for O$_2$ and glucose over distances ranging from 1 μm to 10 m.

| Diffusion distance | Time (10°C) | |
|---|---|---|
| | Oxygen | Glucose |
| 1 μm | 0.34 ms | 1.1 ms |
| 3 μm | 3.1 ms | 10 ms |
| 10 μm | 34 ms | 110 ms |
| 30 μm | 0.31 s | 1.0 s |
| 100 μm | 3.4 s | 10 s |
| 300 μm | 31 s | 100 s |
| 600 μm | 2.1 min | 6.9 min |
| 1 mm | 5.7 min | 19 min |
| 3 mm | 0.8 h | 2.8 h |
| 1 cm | 9.5 h | 1.3 d |
| 3 cm | 3.6 d | 12 d |
| 10 cm | 40 d | 130 d |
| 30 cm | 1.0 yr | 3.3 yr |
| 1 m | 10.8 yr | 35 yr |
| 3 m | 98 yr | 320 yr |
| 10 m | 1090 yr | 3600 yr |

diagenetic models, because the different chemical species have time and length scales of diffusion and reaction which vary over many orders of magnitude and which are therefore correspondingly difficult to compare.

Calculations from Equation 5.8 of the diffusion times of oxygen molecules at 10°C show that it takes one hour for a mean diffusion distance of 3-4 mm, whereas it takes a day to diffuse 2 cm and 1000 years for 10 meters (Table 5.1). For a small organic molecule such as glucose, these diffusion times are about three times longer. Over the scale of a bacterium, however, diffusion takes only 1/1000 second. Thus, for bacteria of 1 μm size, one could hardly envision a transport mechanism which would outrun diffusion within a millisecond. The transition between predominantly diffusive to predominantly advective or turbulent transport of solutes lies in the range of 0.1 mm for actively swimming organisms and somewhat higher for passively sinking marine aggregates (Fig. 5.4).

**5.2.3    Diffusive Boundary Layers**

The transition from a turbulent flow regime with advective and eddy transport to a small scale

dominated by viscosity and diffusional transport is apparent when a solid-water interface such as the sediment surface is approached (Fig. 5.5). According to the classical eddy diffusion theory, the vertical component of the eddy diffusivity, $E$, decreases as a solid interface is approached according to: $E = A \, v \, Z^{3-4}$, where $A$ is a constant, $v$ is the kinematic viscosity, and $Z$ is the height above the bottom. An exponent of 3-4 shows that the eddy diffusivity drops very steeply as the sediment surface is approached. In the viscous sublayer, which is typically =1 cm thick in the deep sea, the eddy diffusivity falls below the kinematic viscosity of ca $10^{-2}$ cm$^2$s$^{-1}$. Even closer to the sediment surface, the vertical eddy diffusion coefficient for mass falls below the molecular diffusion coefficient, D, which is constant for a given solute and temperature, and which for small dissolved molecules is in the order of $10^{-5}$ cm$^2$s$^{-1}$. The level where E becomes smaller than D defines the diffusive boundary layer, $\delta_c$, which is typically about

0.5 mm thick. In this layer, molecular diffusion is the predominant transport mechanism, provided that the sediment is impermeable and stable.

The diffusive boundary layer plays an important role for the exchange of solutes across the sediment-water interface (Jørgensen 1998). For chemical species which have a very steep gradient in the diffusive boundary layer it may limit the flux and thereby the rate of chemical reaction. This may be the case for the precipitation of manganese on iron-manganese nodules (Boudreau 1988) or for the dissolution of carbonate shells and other minerals such as alabaster in the deep sea (Santschi et al. 1991). For chemical species with a weak gradient, such as sulfate, the diffusive boundary layer plays no role since the uptake of sulfate is totally governed by diffusion-reaction within the sediment. The function of the diffusive boundary layer as a barrier for solute exchange is also reflected in the mean diffusion time of molecules through the layer, which is about 1 min over a 0.5 mm distance (Table 5.1).

The existence of a diffusive boundary layer is apparent from microsensor measurements of oxygen and other solutes at the sediment-water interface. In a Danish coastal sediment, the concentration of oxygen dropped steeply over the 0.5 mm thick boundary layer (Fig. 5.5) which consequently had a significant influence on the regulation of oxygen uptake in this sediment of high organic-matter turnover. Figure 5.6 shows, as another example, oxygen penetration down to 13 mm below the surface in a fine-grained sediment . The profile was measured in-situ in the sea bed by a free-falling benthic lander operating with a 100 µm depth resolution, which was just sufficient to resolve the diffusive boundary layer. Based on the boundary layer gradient, the vertical diffusion flux of oxygen across the water-sediment interface can be calculated (cf. Chap. 3).

[Figure]

**Fig. 5.5** Oxygen microgradient (n) at the sediment-water interface compared to the ratio, E/D (logarithmic scale), between the vertical eddy diffusion coefficient, E, and the molecular diffusion coefficient, D. Oxygen concentration was constant in the overflowing seawater. It decreased linearly within the diffusive boundary layer (DBL), and penetrated only 0.7 mm into the sediment. The DBL had a thickness of 0.45 mm. Its effective thickness, $\delta_e$ is defined by the intersection between the linear DBL gradient and the constant bulk water concentration. The diffusive boundary layer occurs where E becomes smaller than D, i.e. where E/D = 1 (arrow). Data from Aarhus Bay, Denmark, at 15 m water depth during fall 1990 (Gundersen et al. 1995).

**5.3   Regulation and Limits of Microbial Processes**

Bacteria and other microorganisms are the great biological catalysts of element cycling at the sea floor. The degradation and remineralization of organic matter and many redox processes among inorganic species are dependent on bacterial catalysis, which may accelerate such processes up to $10^{20}$-fold relative to the non-biological reaction

[Figure]

**Fig. 5.6** Oxygen gradient measured in-situ by a benthic lander in Skagerrak at the transition between the Baltic Sea and the North Sea at 700 m water depth. Due to the high depth resolution of the microelectrode measurements it was possible to analyze the $O_2$ microgradient within the 0.5-mm thick diffusive boundary layer. The framed part in the upper graph is blown up in the lower graph. (Data from Gundersen et al. 1995)

rate. It is, however, important to keep in mind that this biological catalysis is based on living organisms, each of which has its special requirements and limits for its physical and chemical environment, its supply of nutrients, growth rate and mortality, interaction with other organisms, etc..

**5.3.1 Substrate Uptake by Microorganisms**

Among the many reasons for the importance of prokaryotic organisms in biogeochemical cycling are:

A) Their metabolic versatility, especially the many types of anaerobic metabolism and the ability to degrade complex polymeric substances or to catalyze reactions among inorganic compounds.

B) Their small size which allows them to inhabit nearly any environment on the surface of the Earth and which strongly enhances the efficiency of their catalytic activity.

C) The wide range of environmental conditions under which they thrive, including temperature, salinity, pH, hydrostatic pressure, etc.

The metabolic versatility of prokaryotic organisms is discussed in Section 5.4. The small size of the individual cells is related to their nutrition exclusively on solutes such as small organic molecules, inorganic ions or gases. The uptake of food by the prokaryotic cells thus takes place principally by molecular diffusion of small molecules to the cell surface and their transport through the cytoplasmic membrane into the cell. This constrains the relationships between cell size, metabolic rate, substrate concentration and molecular diffusion coefficients (D) (e.g. Koch 1990, 1996, Karp-Boss et al. 1996). The concentration gradient around the spherical cell is:

$$C_r = (R/r) \cdot (C_0 - C_\infty) + C_\infty \quad , r > R \quad (5.10)$$

where $C_r$ is the concentration at the radial distance, $r$, $C_0$ is the substrate concentration at the cell surface, $C_\infty$ is the ambient substrate concentration, and $R$ is the radius of the cell (Fig. 5.7). The maximal substrate uptake rate of a cell is reached when the substrate concentration at the cell surface is zero. The total diffusion flux, $J$, to the cell is then:

$$J = 4 \pi D R C_\infty \quad (5.11)$$

This flux provides the maximal substrate supply to the diffusion-limited cell, which has a volume of $^4/_3 \pi R^3$. The flux thus determines the maximal specific rate of bacterial metabolism of the substrate molecules, i.e. the metabolic rate per volume of biomass:

Specific metabolic rate =
$$(4 \pi D R C_\infty)/(^4/_3 \pi R^3) = (3D/R^2)C_\infty \quad (5.12)$$

Equation 5.12 shows that the biomass-specific metabolic rate of the diffusion-limited cell varies inversely with the square of its size. This means that the cell could potentially increase its specific rate of metabolism 4-fold if the cell diameter were only half as large. The smaller the cell, the less likely it is that its substrate uptake will reach diffusion limitation. Thus, at the low substrate concentrations normally found in marine environments, microorganisms avoid substrate limitation by forming small cells of <1 μm size. Thereby,

the bacteria become limited by their transport efficiency of molecules across the cell membrane rather than by diffusion from their surroundings (Fig. 5.7). In the nutrient-poor seawater, where substrates are available only in sub-micromolar and even nanomolar concentrations, free-living bacteria may have an impressingly high substrate uptake efficiency which corresponds to clearing a seawater volume for substrate each second which is several hundred times their own volume. In sediments, on the other hand, bacteria often form microcolonies or their diffusion supply is impeded by sediment structures so that the microorganisms may at times be diffusion limited in their substrate uptake. It is not clear, however, how this affects the kinetics of substrate turnover in sediments.

The theoretical limit for substrate availability required to sustain bacterial growth and survival is of great biogeochemical significance, e.g. in relation to the exceedingly slow degradation of organic material in million-year old sediments and oil reservoirs deep under the sea floor (Stetter et al. 1993, Parkes et al. 1994). The recent discovery of this 'deep biosphere' has added a new perspective to the limits of life and the regulation of organic carbon burial in marine sediments. It appears, that the gradual geothermal heating of sediments as they become buried to several hundred

meters depth enhances the availability of even highly refractory organic material to microbial attack and leads to further mineralization with the ultimate formation of methane (Wellsbury et al. 1997). The methane slowly diffuses upwards from the deep deposits and becomes oxidized to $CO_2$ as it reaches the bottom of the sulfate zone (Chap. 8), or it accumulates as gas hydrate within the upper sediment strata (Borowski et al. 1996).

**5.3.2    Temperature as a Regulating Factor**

The sea floor is mostly a cold environment with 85% of the global ocean having temperatures below 5°C. At the other extreme, hydrothermal vents along the mid-oceanic ridges have temperatures reaching above 350°C. In each temperature range from the freezing point to an upper limit of around 110°C there appear to be prokaryotic organisms which are well adapted and even thrive optimally at that temperature. Thus, extremely warm-adapted (hyperthermophilic) methane producing or elemental-sulfur reducing bacteria, which live at the boiling point of water, are unable to grow at temperatures below 60-70°C because it is too cold for them (Stetter 1996).

It is well known that chemical processes as well as bacterial metabolism are slowed down by low temperature. Yet, the biogeochemical recycling of deposited organic material in marine sediments does not appear to be less efficient or less complete in polar regions than in temperate or tropical environments. Sulfate reduction rates in marine sediments of below 0°C around Svalbard at 78 degrees north in the Arctic Ocean are comparable to those at similar water depths along the European coast (Sagemann et al. 1998). Figure 5.8A shows the short-term temperature dependence of sulfate reduction rates in such Svalbard sediments. As is typical for microbial processes, the optimum temperature of 25-30°C is high above the in-situ temperature, which was -1.7°C during summer. Above the optimum, the process rate dropped steeply, which is due to enzymatic denaturation and other physiological malfunctioning of the cells and which shows that this is a biologically and not a chemically catalyzed process. Although the sulfate reduction was slow at <0°C this does not mean that the bacteria do not function well at low temperature. On the contrary, the organisms had their highest growth efficiency (i.e. highest biomass production per amount of substrate consumed) at around 0°C, in contrast to

[Figure]

**Fig. 5.7** Theoretical concentration gradient of substrate molecules around a spherical cell at different radial distances from its center (R is the radius of the cell). The concentration of substrate in the bulk water is $C_\infty$ and concentration curves were calculated from Equation 5.10 for a cell limited in its substrate uptake by external diffusion ('Diffusion limited') or by the uptake capacity across its own cell membrane ('Uptake limited'). Note how the substrate concentration only gradually approaches the bulk concentration with increasing distance from the cell.

sulfate reducers from temperate environments which have their highest growth efficiency at the warm temperatures experienced during summer (Isaksen and Jørgensen 1996, C. Knoblauch, unpublished).

The radiotracer method of measuring sulfate reduction in sediments (see Sect. 5.6) is a sensitive tool to demonstrate the temperature strains among the environmental microorganisms. In ca. 100°C warm sediment from the hydrothermal sediments of the Guaymas Basin, Gulf of California, three main groups of sulfate reducers could be discriminated from their temperature optima: a) mesophiles with an optimum at 35°C, b) thermophiles with optimum at 85°C and, c) hyperthermophiles with optimum at 105°C (Fig. 5.8B). The mesophiles comprise the sulfate reducers known best from pure cultures, whereas thermophiles with optimum at 85°C are known among the genus, *Archaeoglobus*. Such prokaryotic organisms have been isolated from hydrothermal environments and have also been found in oil reservoirs 3000 m below the sea bed where the temperature is up to 110°C and the hydrostatic pressure up to 420 bar. Hyperthermophilic sulfate reducing bacteria which can metabolize at >100°C have never been isolated and are still not known.

Surprisingly, sulfate reducers adapted to the normal low temperatures of the main sea floor were also unknown until recently. Such cold-adapted (psychrophilic) bacteria are generally scarce among the culture collections in spite of

their major biogeochemical significance. This is partly because of their slow growth which makes them difficult to isolate and cumbersome to study. A number of psychrophilic sulfate reducers were recently isolated which have temperature optima down to 7°C (C. Knoblauch, unpublished).

These examples demonstrate a general problem in marine microbiology, namely that only a very small fraction (one percent?) of the bacterial species in the ocean are known to science today. Many of the unknown microorganisms may be among the biogeochemically very important species. The estimate is based on recent methodological advances in molecular biology which have made it possible to analyze the diversity of natural prokaryotic populations, even of the many which have still not been isolated or studied.

**5.3.3    Other Regulating Factors**

The major area of the sea floor lies in the deep sea below several thousand meters of water where an enormous hydrostatic pressure prevails. Since the pressure increases by ca 1 bar (1 atm) for every 10 meters, bacteria living at 5000 m depth must be able to withstand a pressure of 500 bar. Microorganisms isolated from sediments down to 3000-4000 m have been found to be preferentially barotolerant, i.e. they grow equally well at sea surface pressure as at their in-situ pressure. At depths exceeding 4000 m the isolated bacteria become increasingly barophilic, i.e. they grow opti-

[Figure]

[Figure]

**Fig. 5.8**  Temperature regulation of bacterial sulfate reduction in different marine sediments. The data show the rates of sulfate reduction in a homogenized sample from the upper 5-10 cm of sediment measured by short-term incubations in a temperature gradient block with radiolabelled sulfate as a tracer. A) Arctic sediment from 175 m depth in Storfjorden on Svalbard at 78°N where the in situ temperature was -1.7°C. B) Sediment from the hydrothermal area of the Guaymas Basin at 2000 m depth. The hydrothermal fluid here seeps up through fine-grained, organic-matter rich sediment which was collected at 15-20 cm depth where the in situ temperature was 90-125°C. (Data from Sagemann et al. 1998, and Jørgensen et al. 1992).

mally at high pressures, and bacteria isolated from deep-sea trenches at 10,000 m depth were found to grow optimally at 700-1000 bar (Yayanos 1986). Barophilic bacteria are also psychrophilic which is in accordance with the low temperature of 2-3°C in the deep sea. They appear to grow relatively slowly which may be an adaptation to low temperature and low nutrient availability rather than a direct effect of high pressure. The degradation of organic material appears to be just as efficient in the deep sea as in shallower water, since only a few percent of sedimenting detritus resist mineralization and are buried deep down into the sediment.

**5.4   Energy Metabolism of Prokaryotes**

Microorganisms can be considered 1 µm large bags of enzymes in which the important biogeochemical processes are catalyzed. This analogy, however, is too crude to understand how and why these microscopic organisms drive and regulate the major cycles of elements in the ocean. This requires a basic knowledge of their energy metabolism and physiology.

The cells use the chemical energy of organic or inorganic compounds for cell functions which require work: growth and division, synthesis of macromolecules, transport of solutes across the cell membrane, secretion of exoenzymes or exopolymers, movement etc. The organisms catalyze redox processes from which they conserve a part of the energy and couple it to the formation of a proton gradient across the cytoplasmic membrane. The socalled *proton motive force* established by this gradient is comparable to the electron motive force of an ordinary battery. It is created by the difference in electrical charge and $H^+$ concentration between the inside (negative and low $H^+$, i.e. alkaline) and the outside (positive and high $H^+$) of the membrane. By reentry of protons into the cell through membrane-bound ATPase protein complexes, it drives the formation of high energy phosphate bonds in compounds such as *ATP* (adenosine triphosphate), which functions as a transient storage of the energy and is continuously recycled as the phosphate bond is cleaved in energy-requiring processes. ATP is utilized very widely in organisms as the fuel to drive energy requiring processes, and a fundamental question in all cellular processes is, how much ATP do they produce or consume?

Reduction-oxidation (redox) processes, whether biological or chemical, involve a transfer of one or more electrons between the chemical reactants. An example is the oxidation of ferrous iron to ferric iron by oxygen at low pH:

$$2Fe^{2+} + \frac{1}{2}O_2 + 2H^+ \rightarrow 2Fe^{3+} + H_2O \qquad (5.13)$$

By this reaction, a single electron is transferred from each $Fe^{2+}$ ion to the $O_2$ molecule. The ferrous iron is thereby oxidized to ferric iron, whereas the oxygen is reduced to water. In the geochemical literature, $O_2$ in such a reaction is termed the *oxidant* and $Fe^{2+}$ the *reductant*. In the biological literature, the terms *electron acceptor* for $O_2$ and *electron donor* for $Fe^{2+}$ are used.

**5.4.1   Free Energy**

Chemical reactions catalyzed by microorganisms yield highly variable amounts of energy and some are directly energy consuming. The term *free energy*, G, of a reaction is used to express the energy released per mol of reactant, which is available to do useful work. The change in free energy is conventionally expressed as $\Delta G^0$, where the symbol $\Delta$ should be read as 'change in' and the superscript $^0$ indicate the following standard conditions: pH 7, 25°C, a 1 M concentration of all reactants and products and a 1 bar partial pressure of gasses. The value of $\Delta G^0$ for a given reaction is expressed in units of kilojoule (kJ) per mol of reactant. If there is a net decrease in free energy ($\Delta G^0$ is negative) then the process is *exergonic* and may proceed spontaneously or biologically catalyzed. If $\Delta G^0$ is positive, the process is *endergonic* and energy from ATP or from an accompanying process is required to drive the reaction. The change in free energy of several metabolic processes in prokaryotic organisms is listed in Table 5.2.

A thorough discussion of the theory and calculation of $\Delta G^0$ for a variety of anaerobic microbial processes is given by Thauer et al. (1977). As a general rule, processes for which the release of energy is very small, < ca 7 kJ mol$^{-1}$, are insufficient for the formation of ATP and are thus unable to serve the energy metabolism of microorganisms. It is important to note, however, that standard conditions are seldom met in the marine environment and that the actual conditions of pH,

**Table 5.2** Pathways of organic matter oxidation, hydrogen transformation and fermentation in the sea floor and their standard free energy yields, $\Delta G^0$, per mol of organic carbon. $\Delta G^0$ values according to Thauer et al. (1977), Fenchel et al. (1998), and Conrad et al. (1986) (cf. Fig.3.11).

| Pathway and stoichiometry of reaction | $\Delta G^0$ (kJ mol-1) |
|---|---|
| **Oxic respiration:** | |
| $CH_2O + O_2 \rightarrow CO_2 + H_2O$ | -479 |
| **Denitrification:** | |
| $5CH_2O + 4NO_3^- \rightarrow 2N_2 + 4HCO_3^- + CO_2 + 3H_2O$ | -453 |
| **Mn(IV) reduction:** | |
| $CH_2O + 3CO_2 + H_2O + 2MnO_2 \rightarrow 2Mn^{2+} + 4HCO_3^-$ | -349 |
| **Fe(III) reduction:** | |
| $CH_2O + 7CO_2 + 4Fe(OH)_3 \rightarrow 4Fe^{2+} + 8HCO_3^- + 3H_2O$ | -114 |
| **Sulfate reduction:** | |
| $2CH_2O + SO_4^{2-} \rightarrow H_2S + 2HCO_3^-$ | -77 |
| $4H_2 + SO_4^{2-} + H^+ \rightarrow HS^- + 4H_2O$ | -152 |
| $CH_3COO^- + SO_4^{2-} + 2H^+ \rightarrow 2CO_2 + HS^- + 2H_2O$ | -41 |
| **Methane production:** | |
| $4H_2 + HCO_3^- + H^+ \rightarrow CH_4 + 3H_2O$ | -136 |
| $CH_3COO^- + H^+ \rightarrow CH_4 + CO_2$ | -28 |
| **Acetogenesis:** | |
| $4H_2 + 2CO_3^- + H^+ \rightarrow CH_3COO^- + 4H_2O$ | -105 |
| **Fermentation:** | |
| $CH_3CH_2OH + H_2O \rightarrow CH_3COO^- + 2H_2 + H^+$ | 10 |
| $CH_3CH_2COO^- + 3H_2O \rightarrow CH_3COO^- + HCO_3^- + 3H_2 + H^+$ | 77 |

temperature and substrate/product concentrations must be known before the energetics of a certain reaction can be realistically calculated. Several processes, which under standard conditions would be endergonic, may be exergonic in the normal marine sediment.

An important example of this is the formation of $H_2$ in several bacterial fermentation processes which is exergonic only under low $H_2$ partial pressure. The hydrogen cycling in sediments is therefore dependent on the immediate consumption of $H_2$ by other organisms, such as the sulfate reducing bacteria, which keep the partial pressure of $H_2$ extremely low. The $H_2$-producing and the $H_2$-consuming bacteria thus tend to grow in close proximity to each other, thereby facilitating the diffusional transfer of $H_2$ at low concentration from one organism to the other, a socalled 'interspecies hydrogen transfer' (Conrad et al. 1986).

**5.4.2 Reduction-Oxidation Processes**

The electron transfer in redox processes is often accompanied by a transfer of protons, $H^+$. The simplest example is the oxidation of $H_2$ with $O_2$ by the socalled 'Knallgas-bacteria', which occur widespread in aquatic sediments:

$$H_2 + \tfrac{1}{2}O_2 \rightarrow H_2O \qquad (5.14)$$

In the energy metabolism of cells, an intermediate carrier of the electrons (and protons) is commonly required. Such an *electron carrier* is, for instance, $NAD^+$ (nicotinamid adenin dinucleotide), which

formally accepts two electrons and one proton and is thereby reduced to NADH. The NADH may give off the electrons again to specialized electron acceptors and the protons are released in the cell sap. Thereby, the NADH, which must be used repeatedly, is recycled.

A redox process such as Equation 5.14 formally consists of two reversible half-reactions. The first is the oxidation of $H_2$ to release electrons and protons:

$$H_2 \Leftrightarrow 2e^- + 2H^+ \qquad (5.15)$$

The second is the reduction of oxygen by the transfer of electrons (and protons):

$$\tfrac{1}{2}O_2 + 2e^- + 2H^+ \Leftrightarrow H_2O \qquad (5.16)$$

Compounds such as $H_2$ and $O_2$ vary strongly in their tendency to either give off electrons and thereby become oxidized ($H_2$) or to accept electrons and thereby become reduced ($O_2$). This tendency is expressed as the redox potential, $E_0'$ of the compounds. This potential is expressed in volts and is measured electrically in reference to a standard compound, namely $H_2$. By convention, redox potentials are expressed for half reactions which are reductions, i.e. 'oxidized form + $e^- \rightarrow$ reduced form'. When protons are involved in the reaction, the redox potential in the biological literature is expressed at pH 7, because the interior of cells is approximately neutral in pH. A reducing compound such as $H_2$, with a strong tendency to give off electrons (Eq. 5.15), has a strongly negative $E_0'$ of -0.414 V. An oxidizing compound such as $O_2$, with a strong tendency to accept electrons (Eq. 5.16), has a strongly positive $E_0'$ of +0.816 V. The free energy yield, $\Delta G^0$, of a process is proportional to the difference in $E_0'$ of its two half-reactions, and the redox potentials of electron donor and electron acceptor in a microbial energy metabolism therefore provide important information on whether they can serve as useful substrates for energy metabolism:

$$\Delta G^0 = -n\,F\,\Delta E_0' \; [\text{kJ mol}^{-1}] \qquad (5.17)$$

where n is the number of electrons transferred by the reaction and $F$ is Faraday's constant (96,485 Coulomb mol$^{-1}$).

The redox potentials of different half-reactions, which are of importance in biogeochemistry, are shown in Figure 5.9. The two 'electron towers' show the strongest reductants (most negative $E_0'$) at the top and the strongest oxidants at the bottom. Reactions between electron donors of more negative $E_0'$ with electron acceptors of more positive $E_0'$ are exergonic and may provide the basis for biological energy metabolism. The larger the drop in $E_0'$ between electron donor and acceptor, the more energy is released. As an example, the oxidation of ferrous iron in the form of $FeCO_3$ with $NO_3^-$ is shown, a process which was discussed in Section 5.1.1 in relation to geochemical pore water gradients in deep-sea sediments (Fig. 5.1, Straub et al. 1996). The oxidation of organic compounds such as lactate or acetate by $O_2$ releases maximum amounts of energy, and aerobic respiration is, accordingly, the basis for metazoan life and for many microorganisms.

**5.4.3    Relations to Oxygen**

Before discussing the biological reactions further, a few basic concepts should be clarified. One is the relation of living organisms to $O_2$. Those organisms, who live in the presence of oxygen, are termed *aerobic* (meaning 'living with oxygen'), whereas those who live in the absence of oxygen are *anaerobic*. This discrimination is both physiologically and biogeochemically important. In all aerobic organisms, toxic forms of oxygen, such as peroxide and superoxide, are formed as by-products of the aerobic metabolism. Aerobic organisms rapidly degrade these aggressive species with enzymes such as catalase, peroxidase or superoxide dismutase. Many organisms, which are obligately anaerobic, lack these enzymes and are not able to grow in the presence of oxygen for this and other reasons. Some organisms need oxygen for their respiration, yet they are killed by higher $O_2$ concentrations. They are *microaerophilic* and thrive at the lower boundary of the $O_2$ zone, between the *oxic* (containing oxygen) and the *anoxic* (without oxygen) environments. Other, non-obligately (facultatively) anaerobic bacteria may live both in *oxidized* and in *reduced* zones of the sediment. The oxidized sediment is characterized by redox potentials, as measured by a naked platinum electrode, of $E_H > 0\text{-}100$ mV and up to about +400 mV. Oxidized sediments are generally brown to olive because iron minerals are in oxidized forms. In most marine sediments the $O_2$ penetration is small relative to the depth of oxidized iron minerals. Most of the oxidized zone is therefore anoxic and has been termed *suboxic* (Froelich et

[Figure]

**Fig. 5.9** The 'electron towers' of redox processes in biogeogeochemistry. By the half-reaction on the left side, electrons are released from an electron donor and are transferred to an electron acceptor in the half-reaction on the right side. The drop in redox potential between donor and acceptor is a measure of the chemical energy released by the process. The redox potentials are here calculated for standard conditions at pH 7 and 1 mM concentrations of substrates and products. As an example, the electron transfer (arrow between electron towers) is shown for the oxidation of ferrous carbonate with nitrate (see text).

al. 1979). The reducing sediment below the sub-oxic zone has $E_H$ below 0-100 mV and down to -200 mV and may be black or gray from different forms of iron sulfide minerals.

**5.4.4    Definitions of Energy Metabolism**

We are now ready to explore the main types of energy metabolism. Microbiologists use a termi-nology for the different types based on three cri-teria: a) the energy source, b) the electron donor and c) the carbon source (Table 5.3). To the term for each of these criteria or their combination is added '-*troph*', meaning 'nutrition'. For each of the three criteria there are two alternatives, thus lead-ing to $2^3 = 8$ possibilities. The combinations are, however, partly coupled and only 4-5 of them are of biogeochemical significance.

**Table 5.3**   Microbiological terminology for different types of energy metabolism.

| Energy source | | Electron donor | | Carbon source | |
|---|---|---|---|---|---|
| Light: | *photo-* | Inorganic: | *litho-* | $CO_2$: | *auto-troph* |
| Chemical: | *chemo-* | Organic: | *organo-* | Organic C: | *hetero-troph* |

The energy source of the *phototrophic* organisms such as plants, algae and photosynthetic bacteria is light. For the *chemotrophic* organisms such as animals, fungi and many bacteria it is chemical energy, e.g. of glucose, methane or ammonium. Many phototrophic microorganisms have the capacity to switch between a phototrophic mode of life in the light and a heterotrophic mode, where they take up dissolved organic substrates or catch prey, or they may do both.

The electron donor of green plants is an inorganic compound, $H_2O$, and the final electron acceptor is $CO_2$:

$$CO_2 + H_2O \rightarrow [CH_2O] + O_2 \qquad (5.18)$$

where $CH_2O$ symbolizes the organic matter in plant biomass. A more accurate stoichiometry, originally suggested by Redfield (1958) for marine phytoplankton assimilating also nitrate and phosphate as nutrients, is:

$$106CO_2 + 16NO_3^- + HPO_4^{2-} + 122H_2O + 18H^+ \rightarrow$$
$$C_{106}H_{263}O_{110}N_{16}P + 138O_2 \qquad (5.18a)$$

which shows that the biomass, in particular due to the lipid fraction, is more reduced than $CH_2$.

The electron donor of many biogeochemically important, phototrophic or chemotrophic bacteria is also inorganic, such as $H_2$, $Fe^{2+}$ or $H_2S$. All these organisms are called *lithotrophic* from 'lithos' meaning rock. In contrast, *organotrophic* organisms such as animals use an organic electron donor:

$$[CH_2O] + O_2 \rightarrow CO_2 + H_2O \qquad (5.19)$$

Finally, organisms such as the green plants, but also many lithotrophic bacteria, are *autotrophs*, i.e. they are able to build up their biomass from $CO_2$ and other inorganic nutrients. Animals and many bacteria living on organic substrates instead incorporate organic carbon into their biomass and are termed *heterotrophs*. Among the prokaryotes, there is not a strict discrimination between autotrophy and heterotrophy. Thus, heterotrophs generally incorporate 4-6% $CO_2$ (mostly to convert $C_3$ compounds to $C_4$ compounds in anaplerotic reactions, in order to compensate for $C_4$ compounds which were consumed for biosynthetic purposes in the cell). This heterotrophic $CO_2$ assimilation has in fact been used to estimate the total heterotrophic metabolism of

bacterioplankton based on their dark $^{14}CO_2$ incorporation. As another example, sulfate reducing bacteria living on acetate are in principle heterotrophs but derive about 30% of their cell carbon from $CO_2$. Aerobic methane oxidizing bacteria, which strictly speaking are also living on an organic compound, incorporate 30-90% $CO_2$.

The three criteria in Table 5.3 can now be used in combination to specify the main types of energy metabolism of organisms. Green plants are photolithoautotrophs while animals are chemoorganoheterotrophs. This detail of taxonomy may seem an exaggeration for these organisms which are in daily terms called photoautotrophs and heterotrophs. However, the usefulness becomes apparent when we need to understand the function of, e.g. chemolithoautotrophs or chemolithoheterotrophs for the cycling of nitrogen, manganese, iron or sulfur in marine sediments (see Sect. 5.5).

**5.4.5 Energy Metabolism of Microorganisms**

The basic types of energy metabolism and representative organisms of each group are compiled in Table 5.4. Further information can be found in Fenchel et al. (1998), Ehrlich (1996), Madigan et al. (1997), and other textbooks. Most photoautotrophic organisms use water to reduce $CO_2$ according to the highly endergonic reaction of Equation 5.18. Since water has a high redox potential it requires more energy than is available in single photons of visible light to transfer electrons from water to an electron carrier, $NADP^+$, which can subsequently reduce $CO_2$ through the complex pathway of the Calvin-Benson cycle. Modern plants and algae, which use $H_2O$ as an electron donor, consequently transfer the electrons in two steps through two photocenters. In photosystem II, electrons are transferred from $H_2O$ which is oxidized to $O_2$. In photosystem I these electrons are transferred to a highly reducing primary acceptor and from there to $NADP^+$. More primitive phototrophic bacteria, which predominated on Earth before the evolution of oxygenic photosynthesis, have only one photocenter and are therefore dependent on an electron donor of a lower redox potential than water. In the purple and green sulfur bacteria or cyanobacteria, $H_2S$ serves as such a low-$E_H$ electron donor. The $H_2S$ is oxidized to $S^0$ and mostly further to sulfate. Some purple bacteria are able to use $Fe^{2+}$ as an electron

donor to reduce $CO_2$. Their existence had been suggested many years ago, but they were discovered and isolated only recently (Widdel et al. 1993, Ehrenreich and Widdel 1994). This group is geologically interesting, since direct phototrophic Fe(II) oxidation with $CO_2$ opens the theoretical possibility for iron oxidation in the absence of $O_2$ some 2.0-2.5 billion years ago, at the time when the great deposits of banded iron formations were formed on Earth.

Some of the purple and green bacteria are able to grow photoheterotrophically, which may under some environmental conditions be advantageous as they cover their energy requirements from light but can assimilate organic substrates instead of spending most of the light energy on the assimilation of $CO_2$. The organisms may either take up or excrete $CO_2$ in order to balance the redox state of their substrate with that of their biomass.

**5.4.6    Chemolithotrophs**

The chemolithotrophs comprise a large and diverse group of exclusively prokaryotic organisms, which play important roles for mineral cycling in marine sediments (Table 5.4). They conserve energy from the oxidation of a range of inorganic compounds. Many are autotrophic and assimilate $CO_2$ via the Calvin cycle similar to the plants. Some are heterotrophic and assimilate organic carbon, whereas some can alter between these carbon sources and are termed *mixotrophs*. Many respire with $O_2$, while others are anaerobic and use nitrate, sulfate, $CO_2$ or metal oxides as electron acceptors.

It is important to note that the autotrophic chemolithotrophs have a rather low growth yield (amount of biomass produced per mol of substrate consumed). For example, the sulfur bacteria use

**Table 5.4**   Main types of energy metabolism with examples of representative organisms.

| Metabolism | | Energy source | Carbon source | Electron donor | Organisms |
|---|---|---|---|---|---|
| Photoautotroph | | Light | $CO_2$ | $H_2O$ | Green plants, algae, cyanobacteria |
| | | | | $H_2S$, $S^0$, $Fe^{2+}$ | Purple and green sulfur bact. (*Chromatium, Chlorobium*), Cyanobacteria |
| Photoheterotroph | | | Org. C $\pm$ $CO_2$ | | Purple and green non-sulfur bact. (*Rhodospirillum, Chloroflexus*) |
| Chemo-lithotroph | Chemolitho-autotroph | Oxidation of inorganic compounds | $CO_2$ | $H_2S$, $S^0$, $S_2O_3^{2-}$, $FeS_2$
$NH_4^+$, $NO_2^-$
$H_2$
$Fe^{2+}$, $Mn^{2+}$ | **Aerobic:**
Colorless sulfur bact. (*Thiobacillus, Beggiatoa*)
Nitrifying bact. (*Thiobacillus, Nitrobacter*)
Hydrogen bact. (*Hydrogenomonas*)
Iron bact. (*Ferrobacillus, Shewanella*) |
| | | | | $H_2 + SO_4^{2-}$
$H_2S/S^0/S_2O_3^{2-} + NO_3^-$
$H_2 + CO_2 \rightarrow CH_4$
$H_2 + CO_2 \rightarrow$ acetate | **Anaerobic:**
Some sulfate reducing bact. (*Desulfovibrio spp.*)
Denitrifying sulfur bact. (*Thiobac. denitrificans*)
Methanogenic bact.
Acetogenic bact. |
| | Mixotroph | | $CO_2$ or Org. C | $H_2S$, $S^0$, $S_2O_3^{2-}$ | Colorless sulfur bact. (some *Thiobacillus*) |
| | Chemolitho-heterotroph | | Org. C | $H_2S$, $S^0$, $S_2O_3^{2-}$
$H_2$ | Colorless sulfur bact. (some *Thiobacillus, Beggiatoa*)
Some sulfate reducing bact. |
| Heterotroph (=chemoorganotroph) | | Oxidation of organic compounds | Org. C (max. 30% $CO_2$) | Org. C | **Aerobic:**
Animals, fungi, many bacteria

**Anaerobic:**
Denitrifying bacteria
Mn- or Fe-reducing bacteria
Sulfate reducing bacteria
Fermenting bacteria |
| | | | Org. $C_1$ (30-90% CO2) | $CH_4$ | Methane oxidizing bact. |

most of the electrons and energy from sulfide oxidation to generate ATP. They need this transient energy storage for $CO_2$ assimilation via the Calvin cycle, which energetically is a rather inefficient pathway in spite of its widespread ocurrence among autotrophic organisms. Only some 10-20% of the electrons derived from $H_2S$ oxidation flow into $CO_2$ and are used for autotrophic growth (Kelly 1982). As a consequence, it has a more limited significance for the overall organic carbon budget of marine sediments whether sulfide is oxidized autocatalytically without bacterial involvement or biologically with a resulting formation of new bacterial biomass (Jørgensen 1987). In spite of this low growth yield, filamentous sulfur bacteria such as *Beggiatoa* or *Thioploca* may often form mats of large biomass at the sediment surface and thereby influence the whole community of benthic organisms as well as the chemistry of the sediment (Fossing et al. 1995).

Among the aerobic chemolithotrophs, colorless sulfur bacteria may oxidize $H_2S$, $S^0$, $S_2O_3^{2-}$ or $FeS_2$ to sulfate. Many $H_2S$ oxidizers, such as *Beggiatoa*, accumulate $S^0$ in the cells and are conspicuous due to the highly light-refractive $S^0$-globules which give *Beggiatoa* mats a bright white appearence. Nitrifying bacteria consist of two groups, those which oxidize $NH_4^+$ to $NO_2^-$ and those which oxidize $NO_2^-$ to $NO_3^-$. The hydrogen oxidizing 'Knallgas-bacteria' mentioned before (Eq. 5.14) oxidize $H_2$ to water. They have a higher efficiency of ATP formation than other chemolithotrophs because their electron donor, $H_2$, is highly reducing which provides a large energy yield when feeding electrons into the respiratory chain (see below). A variety of iron and manganese oxidizing bacteria play an important role for metal cycling in the suboxic zone, but this group of organisms is still very incompletely known. Much research has been done on the important acidophilic forms, e.g. *Thiobacillus ferrooxidans*, associated with acid mine drainage, due to their significance and because it is possible to grow these in a chemically stable medium without exessive precipitation of iron oxides.

The anaerobic chemolithotrophs use alternative electron acceptors such as nitrate, sulfate or $CO_2$ for the oxidation of their electron donor. Thus, several sulfur bacteria can respire with nitrate (denitrifiers) and thereby oxidize reduced sulfur species such as $H_2S$, $S^0$ or $S_2O_3^{2-}$ to sulfate. Also the above mentioned iron oxidizing nitrate reducing bacteria belong in this group. Many

sulfate reducing bacteria can use $H_2$ as electron donor and thus live as anaerobic chemolithotrophs. Two very important groups of organisms are the methanogens and the acetogens. The former, which are archaea and not bacteria, form methane from $CO_2$ and $H_2$:

$$CO_2 + 4H_2 \rightarrow CH_4 + H_2O \qquad (5.20)$$

while the latter form acetate:

$$2CO_2 + 4H_2 \rightarrow CH_3COO^- + H^+ \qquad (5.21)$$

In marine sediments below the sulfate zone, methanogenesis is the predominant terminal pathway of organic carbon degradation. Methane may also be formed from acetate or from organic $C_1$-compounds such as methanol or methylamines:

$$CH_3COO^- + H^+ \rightarrow CH_4 + CO_2 \qquad (5.22)$$

**5.4.7    Respiration and Fermentation**

The best known type of energy metabolism in the sea bed is the aerobic respiration by heterotrophic organisms such as animals and many bacteria. Heterotrophic bacteria take up small organic molecules such as glucose, break them down into smaller units, and ultimately oxidize them to $CO_2$ with oxygen. The first pathway inside the cell, called *glycolysis*, converts glucose to pyruvate and conserves only a small amount of potential energy in ATP (Fig. 5.10). A few electrons are transferred to $NAD^+$ to form reduced NADH. Through the complex cyclic pathway called the *tricarboxylic acid cycle (TCA cycle)*, the rest of the available electrons of the organic substrate are transferred, again mostly to form NADH, and $CO_2$ is released. The electron carrier, NADH is recycled by transferring its electrons via a membrane-bound *electron transport chain* to the *terminal electron acceptor*, $O_2$, which is thereby reduced to $H_2O$. Through the electron transport chain, the proton-motive force across the cell membrane is maintained and most of the energy carrier, ATP, is generated. The energy yield of aerobic respiration is large, and up to 38 mol of ATP may be formed per mol of glucose oxidized. This corresponds to a 43% efficiency of energy utilization of the organic substrate, the rest being lost as entropy (heat). By the transfer of electrons from the electron carrier, NADH, to the terminal electron acceptor, $O_2$, the reduced form of the

carrier, $NAD^+$, is regenerated and the electron flow can proceed in a cyclic manner.

Many heterotrophic prokaryotes are anaerobic and have the ability to use terminal electron acceptors other than $O_2$ for the oxidation of their organic substrates (Fig. 5.10). The denitrifying bacteria respire with $NO_3^-$ which they reduce to $N_2$ and release into the large atmospheric nitrogen pool. The sulfate reducing bacteria can use a range of organic molecules as substrates and oxidize these with the concomitant reduction of $SO_4^{2-}$ to $H_2S$. Some bacteria can use manganese(III, IV) or iron(III) oxides as electron acceptors in their heterotrophic metabolism and reduce these to $Mn^{2+}$ or $Fe^{2+}$.

Many bacteria do not possess a complete TCA-cycle and electron transport chain and are thus not able to use an external electron acceptor as terminal oxidant. This prevents the formation of large amounts of ATP via the electron transport chain and thus leads to a low energy yield of the metabolism. The organisms may still form a small amount of ATP by so-called substrate level phosphorylation through the glycolysis, and this is sufficient to enable growth of these *fermentative* organisms. The cells must, however, still be able

to recycle the reduced electron carrier, NADH, in order to continue the glycolytic pathway. In principle, they do this by transferring the electrons from NADH to an intermediate such as pyruvate ($CH_3COCOO^-$). An example of a fermentation reaction is the formation of lactate by lactic acid bacteria (brackets around the pyruvate indicate that this is an intermediate and not an external substrate being assimilated and transformed):

$$[CH_3COCOO^-] + NADH + H^+ \rightarrow$$
$$CH_3CHOHCOO^- + NAD^+ \qquad (5.23)$$

It is seen that fermentation does not require an external electron acceptor and there is no net oxidation of the organic substrate, glucose. There is rather a reallocation of electrons and hydrogen atoms within the cleaved molecule, whereby a small amount of energy is released. Fermentations often involve a cleavage of the $C_3$ compound to a $C_2$ compound plus $CO_2$. The pyruvate is then coupled to coenzyme-A and cleaved in the form of acetyl-CoA and energy is subsequently conserved by the release of acetate.

$$[CH_3COCOO^-] \rightarrow CH_3COO^- + CO_2 + H_2 \quad (5.24)$$

[Figure]

**Fig. 5.10**   Main pathways of catabolic metabolism in respiring and fermenting heterotrophic organisms (see text).

The $H_2$ is formed by a transfer of electrons from pyruvate via the enzyme ferredoxin to $NAD^+$:

$$NADH + H^+ \rightarrow NAD^+ + H_2 \qquad (5.25)$$

The degradation of organic matter via fermentative pathways to small organic molecules such as lactate, butyrate, propionate, acetate, formate, $H_2$ and $CO_2$ is very important in marine sediments, since these compounds are the main substrates for sulfate reduction and partly for methane formation. A form of inorganic fermentation of sulfur compounds was discovered in recent years in several sulfate reducers and other anaerobic bacteria (Bak and Cypionka 1987). These organisms may carry out a disproportionation of $S^0$, $S_2O_3^{2-}$ or $SO_3^{2-}$ by which $H_2S$ and $SO_4^{2-}$ are formed simultaneously (Eqs. 5.4 and 5.5). Disproportionation reactions have turned out to play an important role in the sulfur cycle of marine sediments and for the isotope geochemistry of sulfur (Jørgensen

1990, Thamdrup et al. 1993, Canfield and Teske 1996).

In comparison to all other heterotrophs, the microorganisms oxidizing methane and other $C_1$ compounds such as methanol, have a unique metabolic pathway which involves oxygenase enzymes and thus requires $O_2$. Only aerobic forms have thus been isolated and studied in laboratory culture, yet methane oxidation in marine sediments is known to take place mostly anaerobically at the deep transition to the sulfate zone. Based on the high activation energy and the apparent requirement for molecular oxygen to attack methane in all known methane oxidizers, it is still questionable whether sulfate reducers may directly oxidize methane (see Sect. 5.1).

Although the different groups of prokaryotes may be catagorized in the scheme of Table 5.4, it should be noted that they are often very diverse and flexible and may thus fit into different categories according to their immediate environmen-

**Table 5.5**   Diversity of sulfur metabolism among the sulfate reducing bacteria. The changes in free energy, $\Delta G^0$, have been calculated for standard conditions. The data show that the disproportionation of elemental sulfur is an endergonic process under standard conditions and therefore requires an efficient removal of the formed $HS^-$ to pull the reaction and make it exergonic. After Cypionka (1994).

| *Pathway* and stoichiometry | $\Delta G^0$ (kJ mol$^{-1}$) |
|---|---|
| *Reduction of sulfur compounds:* | |
| $SO_4^{2-} + 4H_2 + H^+ \rightarrow HS^- + 4H_2O$ | -155 |
| $SO_3^{2-} + 3H_2 + H^+ \rightarrow HS^- + 3H_2O$ | -175 |
| $S_2O_3^{2-} + 4H_2 \rightarrow 2HS^- + 3H_2O$ | -179 |
| $S^0 + H_2 \rightarrow HS^- + H^+$ | -30 |
| *Incomplete sulfate reduction:* | |
| $SO_4^{2-} + 2H_2 + H^+ \rightarrow S_2O_3^{2-} + 5H_2O$ | -65 |
| *Disproportionation:* | |
| $S_2O_3^{2-} + H_2O \rightarrow SO_4^{2-} + HS^- + H^+$ | -25 |
| $4SO_3^{2-} + H^+ \rightarrow 3SO_4^{2-} + HS^-$ | -236 |
| $4S^0 + 4H_2O \rightarrow SO_4^{2-} + 3HS^- + 5H^+$ | +33* |
| *Oxidation of sulfur compounds:* | |
| $HS^- + 2O_2 \rightarrow SO_4^{2-} + H^+$ | -794 |
| $HS^- + NO_3^- + H^+ + H_2O \rightarrow SO_4^{2-} + NH_4^+$ | -445 |
| $S_2O_3^{2-} + 2O_2 + H_2O \rightarrow 2SO_4^{2-} + 2H^+$ | -976 |
| $SO_3^{2-} + {}^1/_2O_2 \rightarrow SO_4^{2-}$ | -257 |

tal conditions and mode of life. A good example are the sulfate reducing bacteria which were previously assumed to be obligate anaerobes specialized on the oxidation of a limited range of small organic molecules with sulfate (Postgate 1979, Widdel 1988). Recent studies have revealed a great diversity in the types of sulfur metabolism in these organisms as listed in Table 5.5 (Bak and Cypionka 1987, Dannenberg et al. 1992, Krekeler and Cypionka 1995). Different sulfate reducing bacteria may, alternatively to $SO_4^{2-}$, use $SO_3^{2-}$, $S_2O_3^{2-}$ or $S^0$ as electron acceptors or may disproportionate these in the absence of an appropriate electron donor such as $H_2$. They may even respire with oxygen or nitrate and may oxidize reduced sulfur compounds with oxygen. There is also evidence from marine sediments and pure cultures that sulfate reducing bacteria may reduce oxidized iron minerals (Coleman et al. 1993) and are even able to grow with ferric hydroxide as the sole electron acceptor (C. Knoblauch, unpublished observations; cf. Sect. 7.4.2.2):

$$8Fe(OH)_3 + CH_3COO^- \rightarrow$$
$$8Fe^{2+} + 2HCO_3^- + 5H_2O + 15OH^- \quad (5.26)$$

**5.5    Pathways of Organic Matter Degradation**

Organic material is deposited on the sea floor principally as aggregates which sink down through the water column as a continuous particle rain (Chap. 4). This particulate organic flux is related to the primary productivity of the overlying plankton community and to the water depth through which the detritus sinks while being gradually decomposed. As a mean value, some 25-50% of the primary productivity reaches the sea floor in coastal seas whereas the fraction is only about 1% in the deep sea (Suess 1980; cf. Fig. 12.1). Within the sediment, most organic material remains associated with the particles or is sorbed to inorganic sediment grains. Freshly arrived material, in particular after the sedimentation of a phytoplankton bloom, often forms a thin detritus layer at the sediment-water interface. This detritus is a site of high microbial activity and rapid organic matter degradation, even in the deep sea where the bacterial metabolism is otherwise strongly limited by the low organic influx (Lochte and Turley 1988). Due to the activities of

detritus-feeding benthic invertebrates and to bioturbation, the particulate organic material is gradually buried into the sediment and becomes an integral part of the sedimentary organic matter.

**5.5.1    Depolymerization of Macromolecules**

The organic detritus is a composite of macromolecular compounds such as structural carbohydrates, proteins, nucleic acids and lipid complexes. Prokaryotic organisms are unable to take up particles or even larger organic molecules and are restricted to a molecular size less than ca 600 daltons (Weiss et al. 1991). A simple organic molecule such as sucrose has a molecular size of 342 daltons. Thus, although some bacteria are able to degrade and metabolize fibers of cellulose, lignin, chitin or other structural polymers, they must first degrade the polymers before they can assimilate the monomeric products (Fig. 5.11). This depolymerization is caused by exoenzymes produced by the bacteria and either released freely into their environment or associated with the outer membrane or cell wall. The latter requires a direct contact between the bacterial cells and the particulate substrate and many sediment bacteria are indeed associated with solid surfaces in the sediment. The excretion of enzymes is a loss of carbon and nitrogen and thus an energy investment of the individual bacterial cells, but these have strategies of optimizing the return of monomeric products (Vetter et al. 1998) and may regulate the enzyme production according to the presence of the polymeric target compound in their environment ('substrate induction'). Thus, there is a positive correlation between the availability of specific polymeric substances in the sediment and the concentration of free enzymes which may degrade them (Boetius and Damm 1998, Boetius and Lochte 1996).

The depolymerization of sedimentary organic matter is generally the rate-limiting step in the sequence of mineralization processes. This may be concluded from the observation, that the monomeric compounds and the products of further bacterial degradation do not accumulate in the sediment but are rapidly assimilated and metabolized. Thus, the concentrations of individual free sugars, amino acids or lipids in the pore water are low. Dissolved organic matter (DOM) is a product of partial degradation and may be released from the sediment to the overlying water. It preferentially consists of complex dissolved polymers and oligomers rather than of monomers. Dissolved organic molecules may not only be taken up by bacteria but may also be removed from the pore water by adsorption onto sediment particles or by condensation reactions, such as monosaccharides and amino acids forming melanoidins (Sansone et al. 1987, Hedges 1978). Organic matter, which is not mineralized, may be adsorbed to mineral surfaces and transformed to 'geomacromolecules' which are highly resistant to enzymatic hydrolysis and bacterial degradation and which become more permanently buried with the sediment (Henrichs 1992, Keil et al. 1994). This buried fraction also contains some of the original organic matter which was deposited as refractory biomolecules such as lignins, tannins, algaenan, cutan and suberan (Tegelaar et al. 1989). The burial of organic matter deep down into the sediment shows a positive correlation with the rate of deposition and constitutes in the order of 5-20% of the initially deposited organic matter in shelf sediments and 0.5-3% in deep-sea sediments (Henrichs and Reeburgh 1987, cf. Fig. 12.1).

**5.5.2    Aerobic and Anaerobic Mineralization**

The particulate detritus consumed by metazoans, and the small organic molecules taken up by microorganisms in the oxic zone, may be mineralized completely to $CO_2$ through aerobic respiration within the individual organisms. The aerobic food chain consists of organisms of very diverse feeding biology and size, but of a uniform type of energy metabolism, namely the aerobic respiration. The oxic zone is, however, generally only mm-to-cm thick in shelf sediments as shown by measurements with $O_2$ microsensors (Chap. 3). In slope and deep-sea sediments the oxic zone expands to many cm or dm depth (Reimers 1987, Glud et al. 1994).

Much of the organic mineralization thus takes place within the anoxic sediment. This anoxic world is inhabited primarily by prokaryotic organisms having a high diversity of metabolic types (Fig. 5.11). There are denitrifiers and metal oxide reducers in the suboxic zone which can utilize a wide range of monomeric organic substances and can respire these to $CO_2$. With depth into the sediment, however, the energy yield of bacterial metabolism becomes gradually smaller and the organisms become narrower in the spectrum of substrates which they can use. While denitrifiers are still very versatile with respect to usable substrates, the sulfate reducers are mostly unable to respire, for example sugars or amino acids. Instead, these monomeric compounds are taken up by fermenting bacteria and converted into a narrower spectrum of fermentation products which include primarily volatile fatty acids such as formate, acetate, propionate and butyrate, as well as $H_2$, lactate, some alcohols and $CO_2$. Through a second fermentation step the products may be focused even further towards the key products, acetate, $H_2$ and $CO_2$. The sulfate reducers depend on these products of fermentation which they can respire to $CO_2$. Several well-known sulfate reducers such as *Desulfovibrio* can only carry out an incomplete oxidation of substrates such as lactate, and they excrete acetate as a product. Other sulfate reducers have specialized on acetate and catalyze the complete oxidation to $CO_2$. The degradative capacity of anaerobic bacteria seems, however, to be broader than formerly expected and new physiological types continue to be discovered. As an example, sulfate reducing bacteria able to oxidize aromatic and aliphatic hydrocarbons have now been isolated, which shows that anaerobic prokaryotes are able to degrade important components of crude oil in the absence of oxygen (Rueter et al. 1994, Rabus et al. 1996).

The methane forming (methanogenic) archaea can use only a narrow spectrum of substrates, primarily $H_2$ plus $CO_2$ and acetate (Eqs. 5.20 and 5.22). Within the sulfate zone, which generally penetrates several meters down into the sea bed, the sulfate reducing bacteria compete successfully with the methanogens for these few substrates and methanogenesis is, therefore, of little significance in the sulfate zone. Only a few 'non-competitive' substrates such as methylamines are used only by the methanogens and not by the sulfate reducers (Oremland and Polcin 1982). Below the sulfate zone, however, there are no available electron acceptors left other than $CO_2$, and methane accumulates here as the main terminal product of organic matter degradation.

**5.5.3    Depth Zonation of Oxidants**

The general depth sequence of oxidants used in the mineralization of organic matter is $O_2 \rightarrow NO_3^- \rightarrow Mn(IV) \rightarrow Fe(III) \rightarrow SO_4^{2-} \rightarrow CO_2$. This sequence corresponds to a gradual decrease

[Figure]

**Fig. 5.11** Pathways of organic carbon degradation in marine sediments and their relation to the geochemical zonations and the consumption of oxidants. (After Fenchel and Jørgensen 1977).

in redox potential of the oxidant (Fig. 5.9) and thus to a decrease in the free energy available by respiration with the different electron acceptors (Table 5.2; cf. Sect. 3.2.5). The $\Delta G^0$ of oxic respiration is the highest, -479 kJ mol$^{-1}$, and that of denitrification is nearly as high. Table 5.2 shows that respiration of one mol of organic carbon with sulfate yields only a small fraction of the energy of respiration with oxygen. The remaining potential chemical energy is not lost but is mostly bound in the product, $H_2S$. This energy may become available to other microorganisms, such as chemo-lithotrophic sulfur bacteria, when the sulfide is transported back up towards the sediment surface and comes into contact with potential oxidants.

The quantitative importance of the different oxidants for mineralization of organic carbon has been studied intensely over the last few decades, both by diagenetic modeling and by incubation experiments. It is generally found that oxygen and sulfate play the major role in shelf sediments, where 25-50% of the organic carbon may be mineralized anaerobically by sulfate reducing bacteria (Jørgensen 1982). With increasing water depth and decreasing organic influx down the continental slope and into the deep sea, the depth of oxygen penetration increases and sulfate reduction gradually looses significance. Nitrate seems to play a minor role as an oxidant of organic matter. Manganese oxides occur in shelf sediments mostly in lower concentrations than iron oxides and, expectedly, the Mn(IV) reduction rates should be lower than those of Fe(III) reduction. Manganese, however, is recycled nearer to the sediment sur-face and relatively faster than iron. These solid-phase oxidants are both dependent on bioturbation as the mechanism to bring them from the sediment surface down to their zone of reaction. The shorter this distance, the faster can the metal oxide be recycled.

Table 5.6 shows an example of process rates in a coastal marine sediment. In this comparison, it is important to keep in mind that the different oxidants are not equivalent in their oxidation capacity. When for example the iron in Fe(III) is reduced, the product is Fe(II) and the iron atoms have been reduced only one oxidation step from +3 to +2. Sulfur atoms in sulfate, in contrast, are reduced eight oxidation steps from +6 in $SO_4^{2-}$ to -2 in $H_2S$. One mol of sulfate, therefore, has 8-fold higher oxidation capacity than one mol of iron oxide. In order to compare the different oxidants and their role for organic carbon oxidation, their reduction rates were recalculated to carbon equivalents on the basis of their change in oxidation step. It is then clear that oxygen and sulfate were the predominant oxidants in this sediment, sulfate oxidizing about 44% of the organic carbon in the sediment. Of the $H_2S$ formed, 15% was buried as pyrite, while the rest was reoxidized and could potentially consume one third of the total oxygen uptake. This redox balance is typical for coastal sediments where up to half of the oxygen taken up by the sediment is used for the direct or indirect reoxidation of sulfide and reduced manganese and iron (Jørgensen 1982).

Manganese and iron behave differently with respect to their reactivity towards sulfide. Iron

Table 5.6    Annual budget for the mineralization of organic carbon and the consumption of oxidants in a Danish coastal sediment, Aarhus Bay, at 15 m water depth. The basic reaction and the change in oxidation step is shown for the elements involved. The rates of processes were determined for one m$^2$ of sediment and were all recalculated to carbon equivalents. From data compiled in Jørgensen (1996).

| Reaction | $\Delta$ Oxidation steps | Measured rate | Estimated carbon equivalents |
|---|---|---|---|
| | | mol m$^{-2}$ yr$^{-1}$ | |
| $CH_2O \rightarrow CO_2$ | C: $\quad 0 \rightarrow +4 = 4$ | 9,9 | 9,9 |
| $O_2 \rightarrow H_2O$ | O: $\quad 2 \cdot 0 \rightarrow -2 = 4$ | 9,2 | 9,2 |
| $NO_3^- \rightarrow N_2$ | N: $\quad +5 \rightarrow 0 = 5$ | 0,15 | 0,19 |
| $Mn(IV) \rightarrow Mn^{2+}$ | Mn: $+4 \rightarrow +2 = 2$ | 0,8 | 0,4 |
| $Fe(III) \rightarrow Fe^{2+}$ | Fe: $+3 \rightarrow +2 = 1$ | 1,6 | 0,4 |
| $SO_4^{2-} \rightarrow HS^-$ | S: $+6 \rightarrow -2 = 8$ | 1,7 | 3,4 |

binds strongly to sulfide as FeS or $FeS_2$, whereas manganese does not. Furthermore, the reduced iron is more reactive when it reaches up near the oxic zone, and $Fe^{2+}$ generally does not diffuse out of the sediment, although it may excape by advective pore water flow (Huettel et al. 1998). The $Mn^{2+}$, in contrast, easily recycles via the overlying water column and can thereby be transported from the shelf out into deeper water. This mechanism leads to the accumulation of manganese oxides in continental slope sediments (see Chap. 11). The important role of manganese and iron in slope sediments is evident from Figure 5.12. Manganese constituted about 5% of the total dry weight of this sediment. Below the manganese zone, iron oxides and then sulfate took over the main role as oxidants. In the upper few cm of the sediment, manganese oxide was the dominant oxidant below the $O_2$ zone (Canfield 1993).

The consecutive reduction of oxidants with depth in the marine sediment and the complex reoxidation of their products constitute the 'redox cascade' (Fig. 5.11). An important function of this sequence of reactions is the transport of electrons from organic carbon via inorganic species back to oxygen. The potential energy transferred from the organic carbon to the inorganic products of oxidation is thereby released and may support a variety of lithotrophic microorganisms. These may make a living from the oxidation of sulfide with Fe(III), Mn(IV), $NO_3^-$ or $O_2$. Others may be involved in the oxidation of reduced iron with Mn(IV), $NO_3^-$

or $O_2$ etc. The organisms responsible for these reactions are only partly known and new types continue to be isolated.

The depth sequence of electron acceptors in marine sediments, from oxygen to sulfate, is accompanied by a decrease in the degradability of the organic material remaining at that depth. This decrease can be formally considered to be the result of a number of different organic carbon pools, each with its own degradation characteristics and each being degraded exponentially with time and depth. This is the 'multi-G model' of Westrich and Berner (1984). The consecutive depletion of these pools leads to a steep decrease in organic carbon reactivity, which follows the sum of several exponential decays, and which can be demonstrated from the depth distribution of oxidant consumption rates. Thus, the $O_2$ consumption rate per volume in the oxic zone of a coastal sediment was found to vary between 3,000 and 30,000 nmol $O_2$ cm$^{-3}$d$^{-1}$ as the oxygen penetration depth varied between 5 mm in winter and 1 mm in summer (Gundersen et al. 1995). Some five cm deeper into the sediment, where sulfate reduction predominated and reached maximal activity, the rates of carbon mineralization decreased 100-fold to 25-150 nmol cm$^{-3}$d$^{-1}$ between winter and summer (Thamdrup et al. 1994). A few meters deep into the sediment, where methanogenesis predominated below the sulfate zone, carbon mineralization rates were again about a 100-fold lower. Although the rates down there may seem insig-

[Figure]

**Fig. 5.12** Depth zonation of reduction rates for the oxidants, Mn(IV), Fe(III) and $SO_4^{2-}$ in a marine sediment from Skagerrak at 700 m water depth. Data from Canfield (1993).

nificant, the slow organic matter decomposition to methane proceeds to great depth in the sediment and is therefore important on an areal basis. This methane diffuses back up towards the sulfate zone where it is oxidized to $CO_2$ at the expense of sulfate (Chap. 8). It may consume 10% or more of the total sulfate reduction in coastal sediments, depending on the depth of sulfate penetration (Iversen and Jørgensen 1985, Niewöhner et al. 1998, Chap. 8).

**5.6    Methods in Biogeochemistry**

A diversity of approaches, of which only a few examples can be discussed here, is used in the study of marine biogeochemistry (Fig. 5.2). Among the important goals is to quantify the rate at which biological and chemical processes take place in different depth zones of the sediment. These are often fast processes for which the reactants may have turnover times in the order of days or hours or even minutes. Data on dissolved species in the pore water and on solid-phase geochemistry can be used in diagenetic models to calculate such rates, as discussed in Chapter 3 (e.g. Schulz et al. 1994). The dynamic process is then derived from the chemical gradients in the pore water and from knowledge about the diffusion coefficient of the chemical species. Such diffusion-diagenesis models work best for either very steep and shallow gradients, e.g. of oxygen for which diffusion is rapid, or very deep gradients, e.g. of sulfate which penetrates deep below the zone affected by biological transport of pore water (bioirrigation) or of solid-phase sediment (bioturbation). For intermediate depth scales, e.g. in the suboxic zone, the burrowing fauna influences the transport processes so strongly that advection and bioirrigation tend to dominate over molecular diffusion. If the transport factor is enhanced to an unknown degree, a rate calculation based on molecular diffusion would be correspondingly wrong. In such cases, it may be more realistic to model the solid phase combined with estimates of the rate of burial and of mixing by the infauna (bioturbation). Burial and mixing rates are most often estimated from the vertical distribution and decay of natural radionuclides such as $^{210}Pb$ in the sediment.

Another problem is that many compounds are not only consumed but also recycled in a chemi-cal zone. Thus, sulfate in the upper sediment layers is both consumed by sulfate reduction and produced by sulfide oxidation so the net sulfate removal may be insignificant relative to the total reduction rate. The net removal of sulfate in the whole sediment column is determined by the amount of sulfide trapped in pyrite etc. and is often only 10% of the gross sulfate reduction. A modeling of the sulfate gradient would in that case lead to a 10-fold underestimate of the total reduction rate. In the upper part of the reducing zone, where rates of sulfate reduction are highest, there may be no net depletion of sulfate and the underestimation would be 100%.

**5.6.1    Incubation Experiments**

To overcome this problem, process rates can be measured experimentally in sediment samples by following the concentration changes of the chemical species with time, either in the pore water or in the solid phase. By such incubation experiments it is critical that the physico-chemical conditions and the biology of the sediment remain close to the natural situation. Since gradual changes with time are difficult to avoid, the duration of the experiment should preferably be restricted to several hours up to a day. This corresponds roughly to one doubling time of normal sediment bacteria. A careful way of containing the sediment samples is to distribute the sediment from different depths into gas-tight plastic bags under an atmosphere of nitrogen and keep these at the natural temperature. The sediment can then be mixed without dilution or contact with a gas phase, and subsamples can be taken over time for the analyses of chemical concentrations. Such a technique has been used to measure the rates of organic matter mineralization from the evolution of $CO_2$ or $NH_4^+$ or to measure the reduction rates of manganese and iron oxides (Fig. 5.12, Canfield et al. 1993).

**5.6.2    Radioactive Tracers**

Radioactive tracers are used in a different manner to identify and measure processes in marine sediments, mostly those carried out by microorganisms, but also purely chemical or physical processes. Radiotracers are mostly applied if chemical analysis alone is too insensitive or if the pathway of processes is more complex or cyclic. Thus, if a process is very slow relative to the pool

**Table 5.7**  Radio-isotopes most often used as tracers in biogeochemistry. The measurement of radioactivity is based on the emission of β-radiation (high-energy electrons) or γ-radiation (electromagnetic) with the specified maximum energy.

| Isotope | Emission mode, $E_{max}$ | Half-life | Examples of application |
|---------|--------------------------|-----------|-------------------------|
| $^3$H    | β, 29 keV   | 12 years   | Turnover of org. compounds, autoradiography |
| $^{14}$C | β, 156 keV  | 5730 years | Turnover of org. compounds, $CO_2$ fixation |
| $^{32}$P | β, 1709 keV | 14 days    | Phosphate turnover & assimilation |
| $^{35}$S | β, 167 keV  | 87 days    | Sulfate reduction, thiosulfate transformations |
| $^{55}$Fe | γ, 6 keV   | 2.7 years  | Iron oxidation and reduction |
| $^{59}$Fe | β, 466 keV | 45 days    | do. |

sizes of the reactants or products, a long-term experiment of many days or months would be required to detect a chemical change. Since this would lead to changing sediment conditions and thus to non-natural process rates, higher sensitivity is required. By the use of a radiotracer, hundred- or thousand-fold shorter experiment durations may often be achieved. An example is the measurement of sulfate reduction rates (see Sect. 5.6.3). Sulfate can be analyzed in pore water samples with a precision of about ±2%, and a reduction of 5-10% or more is, therefore, required to determine its rate with a reasonable confidence. If radioactive, $^{35}$S-labelled sulfate is used to trace the process, it is possible to detect the reduction of only 1/100,000 of the $SO_4^{2-}$ by analyzing the radioactivity of the sulfide formed. The sensitivity by using radiotracer is thus improved >1000-fold over the chemical analysis.

The radioisotopes most often used in biogeochemistry are shown in Table 5.7, together with examples of their application. $^3$H is a weak ß-emitter, whereas $^{32}$P is a hard ß-emitter. $^{14}$C and $^{35}$S have similar intermediate energies. The $^{14}$C is most widely used, either to study the synthesis of new organic biomass through photo- or chemosynthesis ($^{14}CO_2$-assimilation), or to study the transformations and mineralization of organic material such as plankton detritus or specific compounds such as glucose, lactate, acetate and other organic molecules. The $^3$H-labelled substrates have in particular been used in connection with autoradiography, where the incorporation of label is quantified and mapped at high spatial resolution by radioactive exposure of a photographic film emulsion. By this technique it may be possible to demonstrate which microorganisms are actively assimilating the labeled compound. Since phosphorus does not undergo redox processes in sediments, $^{32}$P (or $^{33}$P) is mostly used to study the dynamics and uptake of phosphate by microorganisms. $^{35}$S has been used widely to study processes of the sulfur cycle in sediments, in particular to measure sulfate reduction.

Although radiotracer techniques may offer many advantages, they also have inherent problems. For example, applications of $^{35}$S to trace the pathways of $H_2S$ oxidation have been flawed by isotope exchange reactions between the inorganic reduced sulfur species such as elemental sulfur, polysulfides and iron sulfide (Fossing and Jørgensen 1990, Fossing et al. 1992, Fossing 1995). This isotope exchange means that the sulfur atoms swap places between two compounds without a concomitant net chemical reaction between them. If a $^{35}$S atom in $H_2S$ changes place with a $^{32}$S atom in $S^0$, the resulting change in radioactivity will appear as if $^{35}$S-labelled $H_2S$ were oxidized to $S^0$, although there may be no change in the concentrations of $H_2S$ or $S^0$. When the distribution of $^{35}$S radiolabel is followed with time the results can hardly be distinguished from a true net process and may be incorrectly interpreted as such. There is no similar isotopic exchange with sulfate at normal environmental temperatures which would otherwise prevent its use for the measurement of sulfate reduction rates.

The radioisotopes of iron or manganese, $^{55}$Fe, $^{59}$Fe and $^{54}$Mn, have had only limited application as tracers in biogeochemical studies. Experiments with $^{55}$Fe as a tracer for Fe(III) reduction in marine sediments showed problems of unspecific binding and possibly isotope exchange (King 1983, Roden and Lovley 1993), which has discouraged other researchers from the use of this isotope. Similar problems complicate experiments with manganese as a radiotracer. The iron isotope, $^{57}$Fe, has been used in a completely different manner for the analysis of iron speciation in marine sediments. The isotope, $^{57}$Fe, is added to a sediment and is allowed to equilibrate with the iron species. It can then be used to analyze the oxidation state and the mineralogy of iron by Mössbauer spectroscopy and thus to study the oxidation and reduction of iron minerals.

It has been a serious draw-back in studies of nitrogen transformations that a useful radioisotope of nitrogen does not exist. The isotope, $^{13}$N, is available only at accelerator facilities and has a half-life of 5 min, which strongly limits its applicability. Instead, the stable isotope, $^{15}$N, has been used successfully as a tracer for studies of nitrogen transformations in the marine environment. For example, the use of $^{15}$NO$_3^-$ in a recently developed 'isotope pairing' technique has offered possibilities to study the process of denitrification of $^{15}$NO$_3^-$ to N$_2$ (Nielsen 1992). The $^{15}$NO$_3^-$ is added to the water phase over the sediment and is allowed to diffuse into the sediment and gradually equilibrate with $^{14}$NO$_3^-$ in the pore water. By analyzing the isotopic composition of the formed N$_2$, i.e. $^{14}$N$^{14}$N, $^{14}$N$^{15}$N and $^{15}$N$^{15}$N, it has been possible not only to calculate the rate of denitrification, but even to discriminate between denitrification from nitrate diffusing down from the overlying seawater and denitrification from nitrate

formed by nitrification within the sediment. The results have shown that at the normal, low concentrations of nitrate in seawater, <10-20 μM, the main source of nitrate for denitrification is the internally formed nitrate derived from nitrification.

**5.6.3    Example: Sulfate Reduction**

As an example of a radio-tracer method, the measurement of sulfate reduction rates using $^{35}$SO$_4^{2-}$ according to Jørgensen (1978) and Fossing and Jørgensen (1989) is described in brief (Fig. 5.13). The $^{35}$SO$_4^{2-}$ is injected with a microsyringe into whole, intact sediment cores in quantities of a few microliters which contain ca 100 kBq (1 becquerel = 1 radioactive disintegration per second; 37 kBq = 1 microcurie). After 4-8 hours the core is sectioned, and the sediment samples are fixed with zinc acetate. The zinc a) binds the sulfide and prevents its oxidation and, b) kills the bacteria and prevents further sulfate reduction. The reduced $^{35}$S is then separated from the sediment by acidification in a distillation apparatus, and the released H$_2$$^{35}$S is transferred in a stream of N$_2$ to traps containing zinc acetate where Zn$^{35}$S precipitates. The radioactivities of the Zn$^{35}$S and of the remaining $^{35}$SO$_4^{2-}$ are then analyzed by liquid scintillation counting. In a separate sediment core the concentration gradient of sulfate in the pore water is analyzed, and the rates of sulfate reduction can then be calculated according to the equation:

[Figure]

**Fig. 5.13**   Principle of sulfate reduction measurements in sediments using $^{35}$SO$_4^{2-}$ as a tracer (see text).

Sulfate reduction rate =

$$\phi\left(SO_4^{2-}\right) \cdot \frac{H_2^{35}S}{^{35}SO_4^{2-}} \cdot \frac{24}{t} \cdot 1.06 \text{ nmol } SO_4^{2-}cm^{-3}day^{-1}$$

(5.27)

where $\phi$ is porosity, $(SO_4^{2-})$ is sulfate concentration in the pore water ($\phi\,(SO_4^{2-})$ is then sulfate concentration per volume sediment), $H_2^{35}S$ is radioactivity of total reduced sulfur, $^{35}SO_4^{2-}$ is radioactivity of added sulfate tracer, $t$ is experiment time in hours, and 1.06 is a correction factor for the small dynamic isotope discrimination by the bacteria against the heavier isotope. This formula is a good approximation as long as only a small fraction of the labeled sulfate is reduced during incubation, a condition normally fulfilled in marine sediments.

Sulfate reduction rates in marine sediments commonly lie in the range of 1-100 nmol $SO_4^{2-}$ $cm^{-3}day^{-1}$ (Jørgensen 1982). Since the sulfate concentration in the pore water is around 28 mM or 20-25 $\mu$mol $cm^{-3}$, the turn-over time of the sulfate pool is in the order of 1-100 years. A purely chemical experiment would thus require a month to several years of incubation. This clearly illustrates why a radiotracer technique is required to measure the rate within several hours.

Similar principles as described here are used in measurements of, e.g. the oxidation of $^{14}$C-labelled methane or the formation of methane from $^{14}$C-labelled $CO_2$, acetate or methylamine. Often there are no gaseous substrates or products, and it is not possible to separate the radioisotopes as efficiently as by the measurement of sulfate re-duction or methane formation and oxidation. This is the case for radiotracer studies of intermediates in the mineralization of organic matter in sediments, e.g. of $^{14}$C-labelled sugars, amino acids, or volatile fatty acids, studies which have been important for the understanding of the pathways of organic degradation (Fig. 5.11). These organic compounds, however, occur at low concentrations and have a fast turn-over of minutes to hours, so the sensitivity is less important. More important is the fact that these compounds are at a steady-state between production and consumption, which means that their concentration may not change during incubation in spite of their fast turnover. A chemical experiment would thus not immediately be able to detect their natural dynamics, but a radiotracer experiment may.

**5.6.4    Specific Inhibitors**

This limitation of chemical experiments is often overcome by the use of a specific inhibitor (Oremland and Capone 1988). The principle of an inhibitor technique may be, a) to block a sequence of processes at a certain step in order to observe the accumulation of an intermediate compound, b) to inhibit a certain group of organisms in order to observe whether the process is taken over by another group or may proceed auto-catalytically, c) to let the inhibitor substitute the substrate in the process and measure the transformation of the inhibitor with higher sensitivity. Ideally, the inhibitor should be specific for only the relevant target organisms or metabolic reaction. In reality, most inhibitors have side effects

**Table 5.8**   Some inhibitors commonly used in biogeochemistry and microbial ecology.

| Inhibitor | Process | Principle of function |
|---|---|---|
| BES | Methanogenesis | Blocks $CH_4$ formation (methyl-CoM reductase) |
| $MoO_4^{2-}$ | Sulfate reduction | Blocks $SO_4^{2-}$ reduction (depletes ATP pool) |
| Nitrapyrin | Nitrification | Blocks autotrophic $NH_4^+$ oxidation |
| Acetylene | Denitrification | Blocks N2O → N2 (also blocks nitrification) |
| Acetylene | $N_2$-fixation | C2H2 is reduced to C2H4 instead of N2 → NH4+ |
| DCMU | Photosynthesis | Blocks electron flow between Photosystem II → I |
| Cyanide | Respiration | Blocks respiratory enzymes |
| β-fluorolactate | Lactate metabol. | Blocks heterotrophic metabolism of lactate |
| Chloramphenicol | Growth | Blocks prokaryotic protein synthesis |
| Cycloheximide | Growth | Blocks eukaryotic protein synthesis |

and it is necessary through appropriate control experiments to determine these and to find the minimum inhibitor dose required to obtain the required effect. Some examples are given in Table 5.8.

BES (2-bromoethanesulfonic acid) is a structural analogue of mercaptoethanesulfonic acid, also known as coenzyme-M in methanogenic bacteria, a coenzyme associated with the terminal methylation reactions in methanogenesis. BES inhibits this methylation and thus the formation of methane. It belongs to the near-ideal inhibitors because its effect is specific to the target group of organisms. BES has been used to determine the substrates for methane formation in aquatic sediments and to show that some substrates such acetate and $H_2$ are shared in competition between the methanogens and the sulfate reducers, whereas others such as methylamines are 'non-competitive' substrates which are used by the methanogens alone (Oremland and Polcin 1982, Oremland et al. 1982).

Similarly, molybdate ($MoO_4^{2-}$) together with other group VI oxyanions are analogues of sulfate and inhibit sulfate reduction competitively. Molybdate specifically interferes with the initial 'activation' of sulfate through reaction with ATP and tends to deplete the ATP pool, thus leading to cell death of sulfate reducing bacteria. Also molybdate has been important for clearing up the substrate interactions between methanogens and sulfate reducers in sediments. Molybdate has been used to demonstrate quantitatively which substrates play a role for sulfate reduction in marine sediments (Fig. 5.14). When molybdate is added to a sediment at a concentration similar to that of seawater sulfate, 20 mM, sulfate reduction stops. The organic substrates, which were utilized by the sulfate reducers in the uninhibited sediment and which were kept at a minimum concen-

[Figure]

**Fig. 5.14**  Inhibitor experiment for the demonstration of substrates used by sulfate reducing bacteria in a marine sediment. The concentrations of volatile fatty acids, hydrogen and methane are followed during a time-course experiment over 8 hours. At 3.5 hours (arrow) molybdate was added and the substrates accumulate at a rate corresponding to their rate of consumption before inhibition. The formation of methane shows the release of competition for the common substrates for methanogenesis and sulfate reduction ($H_2$ and acetate). Data from Sørensen et al. (1981).

tration as long as they were active, then suddenly start to accumulate because they are no longer consumed. Since the bacterial processes leading to the formation of these substrates are not inhibited, the substrates will accumulate at a similar rate at which they were used by the sulfate reducers before inhibition. Such experiments have demonstrated that acetate, propionate, butyrate, isobutyrate and $H_2$ are among the most important substrates for sulfate reducing bacteria in marine sediments (Sørensen et al. 1981; Christensen 1984).

Nitrapyrin (N-serve) was first introduced in agriculture as a means to inhibit the conversion of ammonium fertilizer to nitrate with subsequent wash-out of the nitrate. The nitrapyrin blocks the copper-containing cytochrome oxidase involved in the initial enzymatic oxidation of ammonium to hydroxylamine. The rate of ammonium accumulation after nitrapyrin inhibition is thus a measure of the nitrification rate before inhibition (Henriksen 1980).

Acetylene ($C_2H_2$) has been used to study the process of denitrification in which it blocks the last step from $N_2O$ to $N_2$ (Sørensen 1978). It causes an accumulation of $N_2O$ which can be analyzed by a gas chromatograph equipped with an electron capture detector or by a $N_2O$ microelectrode (Revsbech et al. 1988). The $N_2O$ accumulation rate is equal to the denitrification rate before inhibition. Acetylene is also used for studies of $N_2$ fixation. Acetylene has a triple bond analogous to $N_2$ ($HC{\equiv}CH$ vs. $N{\equiv}N$) and can substitute $N_2$ competitively. Instead of reducing $N_2$ to $NH_4^+$, the key enzyme of nitrogen fixation, nitrogenase, reduces acetylene to ethylene ($H_2C{=}CH_2$). The formation rate of ethylene, which is easily analyzed in the headspace by a gas chromatograph with flame ionization detector, is thus a measure of the nitrogen fixation rate.

Other inhibitors are applied in studies of photosynthesis (DCMU) and respiration (cyanide) or are used to distinguish activities of prokaryotic (chloramphenicol) versus eukaryotic (cycloheximide) microorganisms (Table 5.8).

**5.6.5   Other Methods**

Information on the hydrolytic activity in marine sediments has been obtained from the use of model substrates labeled with fluorescent dyes such as methylumbelliferone (MUF) or fluorescein. These substrates may be small dimeric molecules, the hydrolytic cleavage of which releases the fluorescence signal, which is then indicative of the activity of specific enzymes such as glucosidase, chitobiase, lipase, aminopeptidase or esterase (Chrost 1991). Also large fluorescently labeled polymers such as the polysaccharides laminarin or pullulan have been used in experiments to demonstrate the mechanism and kinetics of bacterial degradation (Arnosti 1996).

This is contribution No 254 of the Special Research Program SFB 261 (*The South Atlantic in the Late Quaternary*) funded by the Deutsche Forschungsgemeinschaft (DFG).

**References**

Alperin, M.J. and Reeburgh, W.S., 1985. Inhibition Experiments on Anaerobic Methane Oxidation. Applied and Environmental Microbiology, 50: 940-945.

Arnosti, C., 1996. A new method for measuring polysaccharide hydrolysis rates in marine environments. Organic Geochemistry, 25: 105-115.

Bak, F. and Cypionka, H., 1987. A novel type of energy metabolism involving fermentation of inorganic sulphur compounds. Nature, 326: 891-892.

Benz, M., Brune, A. and Schink, B., 1998. Anaerobic and aerobic oxidation of ferrous iron and neutral pH by chemoheterotrophic nitrate-reduction bacteria. Arch. Microbiology, 169: 159-165.

Berelson, W.M., Hammond, D.E., Smith, K.L. Jr; Jahnke, R.A., Devol, A.H., Hinge, K.R., Rowe, G.T. and Sayles, F. (eds), 1987. In situ benthic flux measurement devices: bottom lander technology. MTS Journal, 21: 26-32.

Berner, R.A., 1980. Early diagenesis: A theoretical approach. Princton Univ. Press, Princton, NY, 241 pp.

Boetius, A. and Lochte, K., 1996. Effect of organic enrichments on hydrolytic potentials and growth of bacteria in deep-sea sediments. Marine Ecology Progress Series, 140: 239-250.

Boetius, A. and Damm, E., 1998. Benthic oxygen uptake, hydrolytic potentials and microbial biomass at the Arctic continental slope. Deep-Sea Research I, 45: 239-275.

Borowski, W.S., Paull, C.K. and Ussler, W., 1996. Marine pore-water sulfate profiles indicate in situ methane flux from underlying gas hydrate. Geology, 24: 655-658.

Boudreau, B.P., 1988. Mass-transport constraints on the growth of discoidal ferromanganese nodules. American Journal of Science, 288: 777-797.

Boudreau, B.P., 1997. Diagenetic models and their impletation: modelling transport and reactions in aquatic sediments. Springer, Berlin, Heidelberg, NY, 414 pp.

Reeburgh, W.S., 1969. Observations of gases in Chesapeake Bay sediments. Limnology and Oceanography, 14: 368-375.

Reimers, C.E., 1987. An in situ microprofiling instrument for measuring interfacial pore water gradients: methods and oxygen profiles from the North Pacific Ocean. Deep-Sea Research, 34: 2019-2035.

Revsbech, N.P., Nielsen, L.P., Christensen, P.B. and Sørensen, J., 1988. Combined oxygen and nitrous oxide microsensor for denitrification studies. Applied and Environmental Microbiology, 54: 2245-2249.

Roden, E.E. and Lovley, D.R., 1993. Evaluation of 55Fe as a tracer of Fe(III) reduction in aquatic sediments. Geomicrobiological Journal, 11: 49-56.

Rueter, P., Rabus, R., Wilkes, H., Aeckersberg, F., Rainey, F.A., Jannasch, H.W. and Widdel, F., 1994. Anaerobic oxidation of hydrocarbons in crude oil by new types of sulphate-reducing bacteria. Nature, 372: 455-458.

Sagemann, J., Jørgensen, B.B. and Greeff, O., 1998. Temperature dependence and rates of sulfate reduction in cold sediments of Svalbard, Arctic Ocean. Geomicrobiological Journal, 15: 83-98.

Sansone, F.J., Andrews, C.C. and Okamoto, M.Y., 1987. Adsorption of short-chain organic acids onto nearshore marine sediments. Geochimica et Cosmochimica Acta, 51: 1889-1896.

Santschi, P.H., Anderson R.F., Fleisher, M.Q. and Bowles, W., 1991. Measurements of diffusive sublayer thicknesses in the ocean by alabaster dissolution, and their implications for the measurements of benthic fluxes. Journal of Geophysical Research, 96: 10.641-10.657.

Schopf, J.W. and Klein, C. (eds), 1992. The proterozoic biosphere. Cambridge University Press, Cambridge, 1348 pp.

Schulz, H.D., Dahmke, A., Schinzel, U., Wallmann, K. and Zabel, M., 1994. Early diagenetic processes, fluxes and reaction rates in sediments of the South Atlantic. Geochimica et Cosmochimica Acta, 58: 2041-2060.

Smith, K.L.Jr., Clifford, C.H. Eliason, A.h., Walden, B., Rowe, G.T. and Teal, J.M., 1976. A free vehicle for measuring benthic community metabolism. Limnology and Oceanography, 21: 164-170.

Sørensen, J., 1978. Denitrification rates in a marine sediment as measured by the acetylene inhibition technique. Applied and Environmental Microbiology, 35: 301-305.

Sørensen, J., Christensen, D. and Jørgensen, B.B., 1981. Volatile fatty acids and hydrogen as substrates for sulfate-reducing bacteria in anaerobic marine sediment. Applied and Environmental Microbiology, 42: 5-11.

Stetter, K.O., Huber, R., Blöchl, E., Knurr, M., Eden, R.D., Fielder, M., Cash, H. and Vance, I., 1993. Hyperthermophilic archaea are thriving in deep North Sea and Alaskan oil reservoirs. Nature, 365: 743-745.

Stetter, K.O., 1996. Hyperthermophilic procaryotes. FEMS Microbiology Revue, 18: 149-158.

Straub, K.L., Benz, M., Schink, B. and Widdel, F., 1996. Anaerobic, nitrate-dependent microbial oxidation of ferrous iron. Applied and Environmental Microbiology, 62: 1458-1460.

Straub, K.L. and Buchholz-Cleven, B.E.E., 1998. Enumeration and detection of anaerobic ferrous iron-oxidizing, nitrate-reducing bacteria from diverse European sediments. Applied and Environmental Microbiology, 64: 4846-4856.

Suess, E., 1980. Particulate organic carbon flux in the oceans-surface productivity and oxygen utilization. Nature, 288: 260-263.

Tegelaar, E.W., de Leeuw, J.W., Derenne, S. and Largeau, C., 1989. A reappraisal of kerogen formation. Geochimica et Cosmochimica Acta, 53: 3103-3106.

Tengberg, A., de Bovee, F., Hall, P, Berelson, W., Chadwick, D., Ciceri, G., Crassous, P., Devol, A., Emerson, s., Gage, J., Glud, R., Graziottin, F., Gundersen, J., Hammond, D., Helder, W., Hinga, K., Holby, O., Jahnke, R., Khripounoff, A., Lieberman, S., Nuppenau, V., Pfannkuche, O., Reimers, C., Rowe, G., Sahami, A., Sayles, F., Schurter, M., Smallman, D., Wehrli, B. and de Wilde, P., 1995. Benthic chamber and profiling landers in oceanography - A review of design, technical solutions and function. Progress in Oceanography, 35: 253-292.

Thamdrup, B., Finster, K., Hansen, J.W. and Bak, F., 1993. Bacterial disproportionation of elemental sulfur coupled to chemical reduction of iron or manganese. Applied and Environmental Microbiology, 59: 101-108.

Thamdrup, B., Fossing, H. and Jørgensen, B.B., 1994. Manganese, iron, and sulfur cycling in a coastal marine sediment, Aarhus Bay, Denmark. Geochimica et Cosmochimica Acta, 58: 5115-5129.

Thauer, R.K., Jungermann, K. and Decker, K., 1977. Energy conservation in chemotrophic anaerobic bacteria. Bacterial Reviews, 41: 100-180.

Thomsen, L., Jähmlich, S., Graf, G., Friedrichs, M., Wanner, S. and Springer, B., 1996. An instrument for aggregate studies in the benthic boundary layer. Marine Geology, 135: 153-157.

Vetter, Y.A., Deming, J.W., Jumars, P.A. and Kriegerbrockett, B.B., 1998. A predictive model of bacterial foraging by means of freely released extracellular enzymes. Microbiology Ecology, 36: 75-92.

Weiss, M.S., Abele, U., Weckesser, J., Welte, W. und Schulz, G.E., 1991. Molecular architecture and electrostatic properties of a bacterial porin. Science, 254: 1627-1630.

Wellsbury, P., Goodman, K., Barth, T., Cragg, B.A., Barnes, S.P. and Parkes R.J., 1997. Deep marine biosphere fuelled by increasing organic matter availability during burial and heating. Nature, 388: 573-576.

Westrich, J.T. and Berner, R.A., 1984. The role of sedimentary organic matter in bacterial sulfate reduction: The G model tested. Limnology and Oceanography, 29: 236-249.

Widdel, F., 1988. Microbiology and ecology of sulfate-and sulfur-reduction bacteria. In: Zehnder, A.J.B. (ed). Biology of anaerobic microorganisms. Wiley & Sons, NY, 469-585 pp.

Widdel, F., Schnell, S., Heising, S., Ehrenreich, A., Assmus, B. and Schink, B., 1993. Ferrous iron oxidation by anoxygenic phototrophic bacteria. Nature, 362: 834-836.

Yayanos, A.A., 1986. Evolutional and ecological implications of the properties of deep-sea barophilic bacteria. Proc. Natl. Acad. Sci., 83: 9542-9546.